# Auto-Comp: An Automated Pipeline for Scalable Compositional Probing of Contrastive Vision-Language Models

## Abstract

Modern Vision-Language Models (VLMs) exhibit a critical flaw in compositional reasoning, often confusing "a red cube and a blue sphere" with "a blue cube and a red sphere". Disentangling the visual and linguistic roots of these failures is a fundamental challenge for robust evaluation. To enable fine-grained, controllable analysis, we introduce Auto-Comp, a fully automated and synthetic pipeline for generating scalable benchmarks. Its controllable nature is key to dissecting and isolating different reasoning skills. Auto-Comp generates paired images from Minimal (e.g., "a monitor to the left of a bicycle on a white background") and LLM-generated Contextual captions (e.g., "In a brightly lit photography studio, a monitor is positioned to the left of a bicycle"), allowing a controlled A/B test to disentangle core binding ability from visio-linguistic complexity. Our evaluation of 20 VLMs on novel benchmarks for color binding and spatial relations reveals universal compositional failures in both CLIP and SigLIP model families. Crucially, our novel "Confusion Benchmark" reveals a deeper flaw beyond simple attribute swaps: models are highly susceptible to low-entropy distractors (e.g., repeated objects or colors), demonstrating their compositional failures extend beyond known bag-of-words limitations. we uncover a surprising trade-off: visio-linguistic context, which provides global scene cues, aids spatial reasoning but simultaneously hinders local attribute binding by introducing visual clutter. We release the Auto-Comp pipeline to facilitate future benchmark creation, alongside all our generated benchmarks (`https://huggingface.co/AutoComp`).

## 1 Introduction

Contrastive Vision-Language Models (VLMs) have demonstrated remarkable capabilities in connecting text and images, enabling applications ranging from detailed image captioning to segmentation and image generation (Radford et al., 2021; Ramesh et al., 2022; Rao et al., 2022). By learning from vast, web-scale datasets, these models build powerful, unified embedding spaces for images and text. However, a critical failure point persists in their ability to perform robust compositional reasoning, the capacity to understand and correctly bind elements within a scene based on their attributes and relationships. For instance, a state-of-the-art VLM may correctly identify the presence of a "red cube" and a "blue sphere" in an image, yet fail to distinguish it from an image containing a "blue cube" and a "red sphere". This failure in compositional binding is not a minor flaw; it reveals a fundamental weakness in achieving true, human-like scene understanding and limits the reliability of these models in high-stakes applications.

Evaluating this compositional capability is notoriously difficult. Benchmarks built on real-world images (Krishna et al., 2017; Ma et al., 2023; Hsieh et al., 2023; Dumpala et al., 2024), while ecologically valid, are inherently noisy, making it impossible to isolate specific reasoning failures from confounding visual variables. Conversely, existing synthetic benchmarks that offer more control often lack the photorealism needed to fairly evaluate modern VLMs or use simplistic, template-based language that fails to mirror real-world linguistic diversity (Rizzoli et al., 2025). This leaves a critical question unanswered: do these models have a core compositional ability that is simply brittle to the *visio-linguistic complexity* of realistic scenes, or is their understanding fundamentally flawed?

Consequently, a significant gap exists for a diagnostic tool that can systematically disentangle these factors.

To solve this disentanglement challenge, we introduce **Auto-Comp**, a fully automated pipeline for generating scalable, photorealistic benchmarks. The core innovation of our framework is a parallel generation process designed specifically for controlled A/B testing. For each compositional concept, Auto-Comp creates two distinct visio-linguistic conditions: (1) a *Minimal* condition, featuring simple, template-based captions and objects isolated on a white background to test a model's core binding ability; and (2) a *Contextual* condition, where an LLM rewrites the caption into natural language and a text-to-image model renders the objects within a realistic scene. This parallel structure allows us to precisely measure how performance changes when moving from a sterile to a complex environment. To ensure data quality, the entire pipeline incorporates a rigorous two-stage validation process and is designed to be general, enabling the generation of benchmarks for any user-defined compositional skill.

Through this work, we make the following contributions:

- We present **Auto-Comp**, a fully automated and scalable pipeline for generating and validating high-quality, photorealistic synthetic datasets, enabling the creation of custom evaluation data for any user-defined compositional skill.

- We propose a novel A/B evaluation framework using parallel *Minimal* and *Contextual* visio-linguistic conditions to conduct a controlled study of how complexity impacts a VLM's compositional reasoning.

- Using Auto-Comp, we generate and release Auto-Comp-CP, comprising two comprehensive benchmarks for *Color* (N=1, 2, 3) and *Position* (N=2, 3) binding. For compositional tasks (N≥2), we introduce two novel hard-negative evaluation schemes: the *Swap Benchmark* and the more challenging *Confusion Benchmark*.

- We conduct an extensive analysis of over 20 VLMs, revealing a clear performance hierarchy where SigLIP models outperform CLIP and uncovering a surprising trade-off: visio-linguistic context aids spatial reasoning but hinders attribute binding.

Our work provides an automatic, powerful and scalable new set of tools for the community, enabling a deeper understanding of current VLM limitations, paving the way for the development of more robust models, and providing a flexible tool for future investigations into other compositional skills.

Table 1: Comparison of compositional VLM benchmarks. Our work, Auto-Comp, is the first to combine photorealistic images with a fully automatic, concept-driven generation pipeline.

| Benchmark | # of Samples | Realistic Images | Realistic Text | Concept-driven Generation | Fully Automatic |
|---|---|---|---|---|---|
| CREPE (Ma et al., 2023) | 370,000+ | ✓ | ✗ | ✗ | ✓ |
| SugarCrepe (Hsieh et al., 2023) | 7,512 | ✓ | ✓ | ✗ | ✗ |
| SugarCrepe++ (Hsieh et al., 2023) | 4,757 | ✓ | ✓ | ✗ | ✗ |
| Winoground (Thrush et al., 2022) | 400 | ✓ | ✓ | ✗ | ✗ |
| CIVET (Rizzoli et al., 2025) | ∼35,000 | ✗ | ✓ | ✓ | ✓ |
| **Auto-Comp-CP (Ours)** | **175,221** | ✓ | ✓ | ✓ | ✓ |

## 2 RELATED WORK

Our research is situated at the intersection of VLM evaluation, compositional reasoning, and synthetic data generation. We position our contributions in relation to the key paradigms of prior work, summarized in Table 1.

**Data-driven Benchmarks on Real Images.** A primary approach to compositional VLM evaluation uses pre-existing datasets, where innovation focuses on curating the negative texts. The large-scale CREPE benchmark (Ma et al., 2023), for instance, starts from real image-caption pairs and programmatically swaps attributes to create hard negatives. While automated, this process can produce linguistically awkward captions, creating an artifact that models may exploit. Addressing this, SugarCrepe (Hsieh et al., 2023) and its successor, SugarCrepe++ (Dumpala et al., 2024), leverage LLMs to generate fluent, realistic negative captions, significantly improving text quality.

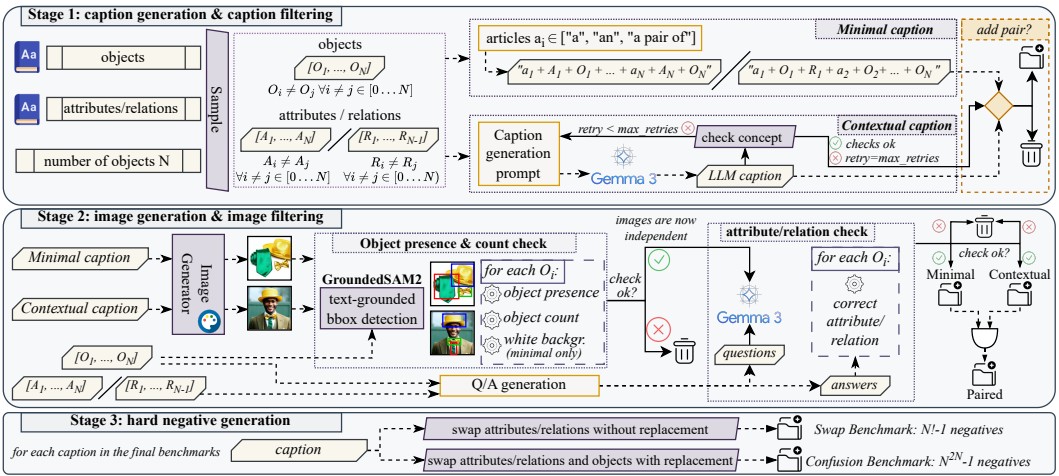

Figure 1: The Auto-Comp pipeline for automated benchmark creation. (1) Caption Generation: Concept-driven generation of template-based Minimal and LLM-based Contextual captions. (2) Image Synthesis & Validation: A text-to-image model generates visuals, which are then validated using GroundingDINO and VLM checks. (3) Hard Negative Generation: Validated pairs are systematically altered by swapping attributes to create challenging compositional test sets.

However, despite achieving realism in both modalities, these benchmarks required human evaluation and remain fundamentally data-driven, not concept-driven. They are constrained by the content of pre-existing images and thus cannot control and isolate objects and distracting elements.

**Concept-driven Benchmarks with Synthetic Visuals.** To gain composition control, a second line of research uses synthetic visuals. The CIVET (Rizzoli et al., 2025) framework is a prime example, programmatically placing real-object cutouts onto a simple grid to create scenes. While this methodology offers precise, concept-level control, it comes at the cost of photorealism. The resulting simplistic, composed scenes create a significant domain gap with the complex, organic images used to train modern VLMs. This gap calls into question the evaluation's relevance, as model performance on such synthetic data may not translate to real-world compositional understanding.

**Our Contribution: Unifying Realism and Concept-driven Generation.** Our work is the first to unify these desired properties. We introduce **Auto-Comp**, a framework that pairs the concept-driven control of synthetic benchmarks with the photorealism of data-driven approaches. As summarized in Table 1, our pipeline fully automatically generates high-fidelity images and realistic text from any user-defined concept. Crucially, our framework is the first to introduce the parallel generation of *Minimal* and *Contextual* visio-linguistic conditions for each concept. This parallel construction provides a unique diagnostic tool for measuring the performance gap between a model's foundational binding capacity in sterile environments and its practical robustness in complex, realistic scenes.

## 3 METHODOLOGY

Our methodology is a fully automated, concept-driven pipeline for generating and validating photorealistic benchmarks at scale. As illustrated in Figure 1, the process consists of three main phases: concept definition, parallel generation, and automated validation.

**Concept Definition.** The foundation of our pipeline is the *Concept*, a structured definition of a scene's intended ground truth. A concept $C$ is formally defined as a tuple $C = (\mathcal{O}_C, \mathcal{A}_C)$, where $\mathcal{O}_C = \{o_1, \ldots, o_N\}$ is a set of $N$ objects and $\mathcal{A}_C$ is a set of attributes. The structure of $\mathcal{A}_C$ is task-dependent: for *Color Binding*, it contains $N$ colors with a one-to-one mapping to objects, while for *Position Binding*, it contains $N-1$ binary spatial relations that chain the objects in sequence. The parameter $N$ controls compositional complexity, with $N = 1$ serving as a baseline and $N \geq 2$ used to evaluate true compositional binding. Vocabularies are detailed in Section 4.

**Phase 1: Parallel Caption Generation.** For each concept, our pipeline generates two distinct styles of captions in parallel. This dual-track generation, designed to represent the same ground-truth with different visio-linguistic properties, is the foundation of our A/B testing framework.

*Minimal Captions.* This track produces a direct, structured description of the concept using a pro-grammatic, template-based approach. The template joins the object-attribute pairs into a factual list, handling grammatical nuances (e.g., articles, pluralization) and adding the suffix "on a white background". The resulting caption serves to test a model's core binding ability in an isolated, unambiguous setting.

*Contextual Captions.* This track produces a fluent, naturalistic description of the same concept. The corresponding concept is provided to a generative LLM (Gemma3-12b-it (Team et al., 2025)) with a carefully engineered few-shot prompt that defines its role and provides instructions and ex-amples. The full prompt is provided in the appendix. To guarantee the LLM's adherence to these instructions, we perform an automated *semantic preservation check* immediately after generation. This check programmatically parses the LLM's output to verify that all original objects and their assigned attributes from the concept $C$ are present and correctly associated. We define a "correct association" with a strict syntactic pattern:

- **For Color Binding,** given an object-color pair $(o_i, a_i)$, the generated text must contain a substring that matches the structure:

$$\langle \texttt{Article} \rangle \ [\langle \texttt{Word} \rangle] \ a_i \ o_i$$

  This pattern ensures the color attribute $a_i$ immediately precedes the object $o_i$, optionally separated by a single modifier word (e.g., "*an elegant red chair*").

- **For Position Binding,** given a chained relation $(o_i, r_i, o_{i+1})$, the generated text must match the pattern:

$$\langle \texttt{Article} \rangle \ [\langle \texttt{Word} \rangle] \ o_i \ r_i \ \langle \texttt{Article} \rangle \ [\langle \texttt{Word} \rangle] \ o_{i+1}$$

  This ensures the relation $r_k$ correctly links the two objects, while allowing for an optional modifier word before each.

If the LLM's output fails to match these patterns for all required bindings after several retries, the entire concept is discarded. This means both the failed Contextual Caption and its successfully generated Minimal Caption are removed. This strict pairing protocol guarantees a clean, one-to-one correspondence for every concept that proceeds to the image generation phase.

**Phase 2: Image Synthesis and Automated Validation.** From this point forward, the Minimal and Contextual tracks are processed independently to maximize the final dataset sizes. For each caption, we generate an image using StableDiffusion3.5-large (Esser et al., 2024b). Recognizing that generative models themselves can fail at composition, we introduce a rigorous, two-stage validation pipeline for every generated image to ensure its fidelity.

*Object Presence and Count Check.* The first, faster validation stage verifies that all objects from the source concept $C$ are present in the image in their correct number. We employ GroundedSAM2 (Ren et al., 2024; Ravi et al., 2024), an open-vocabulary detection model, querying it with each object's class name to handle both singular and plural nouns (e.g., ensuring "gloves" corresponds to two instances). Formally, this validation, $V_{\text{obj}}(G(c), C)$, passes only if the detected objects perfectly match the concept's objects $\mathcal{O}_C$ in both class and cardinality.

*Background Check for Minimal Images.* For images in the Minimal track, we perform an additional validation step. Using the bounding boxes from the object check, we mask all objects and analyze the remaining pixels to verify the background is uniformly white. This check, detailed in the Appendix, allows for minor lighting variations and shadows inherent to the generative process, ensuring each image adheres to the "on a white background" clause and provides a truly sterile visual environment.

*Attribute Correctness Check.* For an image $I \in \mathcal{I}'_{\text{min}} \cup \mathcal{I}'_{\text{ctx}}$ that has passed the object check, we next verify its attributes. We programmatically generate direct questions based on the source concept $C$ (e.g., "*What color is the monitor?*") and leverage the Gemma-3-12b-it model (Team et al., 2025) to provide the answer. To prevent ambiguity, we constrain the VLLM's response to our predefined attribute vocabularies. Formally, we define this validation function as $V_{\text{attr}}(I, C)$, which returns true only if the VLLM provides the correct answer for all attribute questions derived from $C$.

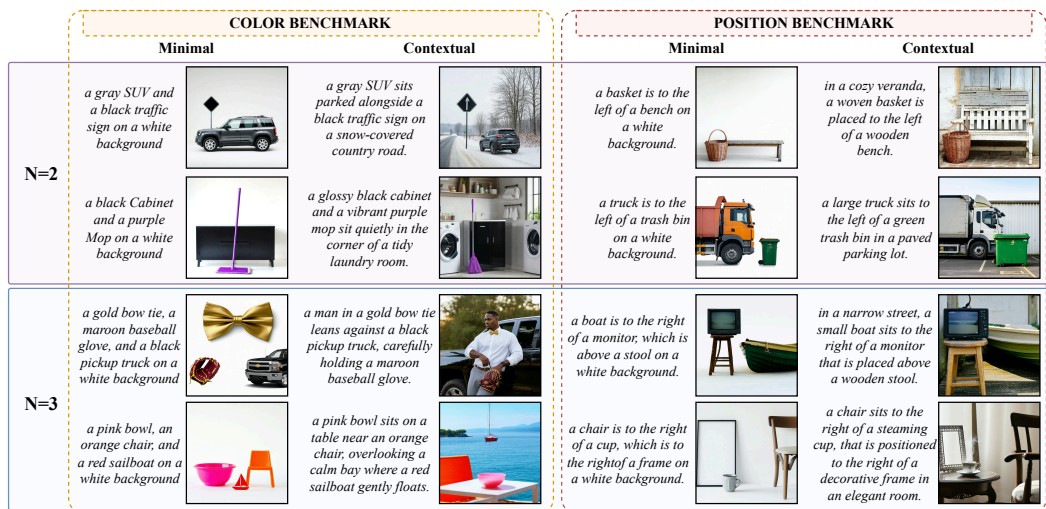

Figure 2: Examples of positive captions-image pairs from the Auto-Comp-CP benchmarks.

To ensure the reliability of the proposed image-validation approach, we first conducted a comprehensive human evaluation study (detailed in Section 6) to validate our automated judge. This study established that our VLLM achieves over 94% concordance with expert human annotators, validating its use as a scalable and highly accurate proxy for human judgment.

*Positive Benchmark.* The positive benchmarks are curated from samples that successfully pass both validation stages. Our large, standalone benchmarks, $\mathcal{B}_{\min}$ and $\mathcal{B}_{\text{ctx}}$, are composed of all validated image-caption pairs for the Minimal and Contextual tracks, respectively. Finally, our *Paired Comparison Set*, $\mathcal{B}_{\text{paired}}$, is created by taking the intersection of these two sets based on their source concepts. This set contains only concepts for which *both* the minimal and contextual versions were successfully validated, enabling a perfectly controlled A/B test. We show qualitative examples in Figure 2. The full benchmark is available in the repository linked in the Abstract and Appendix.

**Phase 3: Hard Negative Benchmark Generation**   The final phase of our pipeline programmatically generates hard negative captions. Unlike prior work that modifies complex, real-world captions, our pipeline begins with text whose structure has been programmatically generated and verified. This high degree of control is a key strength. Unlike LLM-based rewriting, which can inadvertently alter sentence structure, our programmatic swapping ensures positive and negative captions are perfectly controlled, identical in length and complexity, differing *only* in the swapped conceptual elements. Our pipeline automatically handles all grammatical details to maintain linguistic naturalness. As we will demonstrate with a blind LLM evaluation (Section 6), this process creates negatives that are free of artifacts, forcing models to reason compositionally. For each concept with $N \geq 2$ objects, we generate two distinct hard negative benchmarks:

*The Swap Benchmark.* This benchmark is a direct test of compositional binding. Our framework handles two general cases for creating swap-based negatives. For *Attribute Binding Tasks* involving properties like color, we generate negatives by permuting the $N$ attributes among the $N$ fixed objects. For *Relational Binding Tasks* (e.g., spatial position), where permuting the relations is nonsensical for $N = 2$, we instead create foils by permuting the $N$ objects themselves (e.g., "*a table on top of a chair*" from "*a chair on top of a table*"). In either case, this process generates $N! - 1$ unique hard negatives from the original positive, directly measuring a model's ability to associate each element with its correct role.

*The Confusion Benchmark.* To move beyond simple binding errors, we design the Confusion Benchmark, a more strenuous test intended to probe for reliance on brittle heuristics and susceptibility to low-entropy distractors. Here, we generate an exhaustive set of captions by creating all possible combinations of objects and attributes from the original concept, *with replacement*. The generation process depends on the attribute type. For an attribute like color, where each of the $N$ objects is assigned one of $N$ colors, we generate all $N^N$ object arrangements and all $N^N$ color arrangements.

Table 2: Final number of validated samples in each benchmark.

| Benchmark | # Minimal | # Contextual | # Paired |
|---|---|---|---|
| Color N=1 | 16,325 | 8,505 | 4,387 |
| Color N=2 | 26,388 | 36,200 | 15,274 |
| Color N=3 | 3,074 | 2,930 | 1,027 |
| Position N=2 | 38,393 | 30,694 | 17,345 |
| Position N=3 | 7,478 | 5,234 | 1,201 |
| **Total** | **91,658** | **83,563** | **39,234** |

Table 3: Sample survival rates (%) for each benchmark after each validation stage.

| Benchmark | Minimal | | | Contextual | |
|---|---|---|---|---|---|
| | Obj. Check | BG Check | Attr. Check | Obj. Check | Attr. Check |
| Color N=1 | 28.6% | 24.9% | 16.3% | 16.6% | 8.5% |
| Color N=2 | 74.1% | 67.6% | 26.4% | 79.5% | 36.2% |
| Color N=3 | 33.6% | 29.1% | 3.1% | 35.7% | 2.9% |
| Position N=2 | 78.8% | 65.2% | 38.4% | 75.4% | 30.7% |
| Position N=3 | 49.2% | 31.6% | 7.5% | 43.3% | 5.2% |

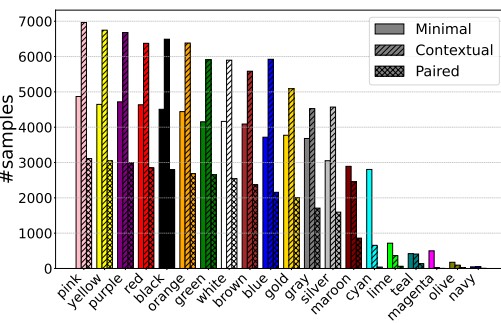 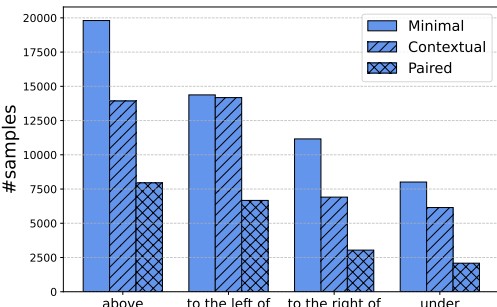

Figure 3: Distribution of validated samples per attribute in our final datasets, summed across N=2 and N=3 tasks. (Left) Total samples per color in the Color Benchmark. (Right) Total samples per relation in the Position Benchmark.

The Cartesian product yields $N^{2N} - 1$ hard-negative captions. For position, which describes the $N - 1$ relative relations between the $N$ objects, the calculation is different. We again generate all $N^N$ object arrangements. However, we generate all possible arrangements for the $N - 1$ relation slots, sampling from the $N - 1$ available position types. This results in $(N - 1)^{N-1}$ unique positional arrangements, producing a total of $N^N(N - 1)^{N-1} - 1$ hard negatives. This benchmark creates a massive set of challenging distractors, allowing us to quantify a model's robustness.

## 4 THE AUTO-COMP-CP BENCHMARK SUITE

**Vocabularies.** Our Auto-Comp-CP (Color-Position) benchmark relies on 3 starting vocabularies.

*Object Vocabulary.* To ensure diversity and clarity, we curated our object vocabulary from the 365 classes in the Object365 dataset Shao et al. (2019). We filtered out ambiguous concepts, abstract nouns, and non-object categories (e.g., "person," animals) to avoid common model biases, resulting in a final vocabulary of 250 distinct objects.

*Color Vocabulary.* We defined a vocabulary of 20 common colors, built upon standard VGA colors (e.g., red, blue, green, yellow, magenta) and supplemented with common shades (e.g., gold, brown) to create challenging evaluation samples with similar yet distinct colors.

*Spatial Relation Vocabulary.* For relational binding, we selected a set of 4 fundamental binary spatial relations inspired by prior work Rizzoli et al. (2025): *over*, *under*, *to the left of*, and *to the right of*. These are used to chain objects together in sequence.

**Dataset Statistics and Analysis.** We generate $100k$ pairs of Minimal/Contextual captions per benchmark in Stage 1 (excluding N=1 color benchmark, for which we start from $20k$ as we use it only for ablation). After Stage 2 and 3, our pipeline yields a large-scale benchmark suite, with over 175,000 validated samples generated across the standalone benchmarks ( Table 2). The validation survival rates, detailed in Table 3, allow us to pinpoint the specific failure modes of the generative model. While the initial *Object Presence Check* filters many samples and the subsequent *Background Check* further purifies the Minimal set, the most significant drop in survival rate across both tracks consistently occurs at the final *Attribute Correctness Check*. This reveals that true compositional binding is the primary bottleneck even for the generative model. Notably, the model struggles

Table 4: Full model performance on Color and Position benchmarks, results in percent (%). **Abbreviations**: Min=Minimal, Ctx=Contextual; S=SigLIP, S2=SigLIP2; OC=OpenCLIP; P=Patch size.

| | | Color Benchmark | | | | | | | | Position Benchmark | | | | | | | |
| | | N=2 Objects | | | | N=3 Objects | | | | N=2 Objects | | | | N=3 Objects | | | |
| | | Min | | Ctx | | Min | | Ctx | | Min | | Ctx | | Min | | Ctx | |
| Family | Model | Swap | Conf. | Swap | Conf. | Swap | Conf. | Swap | Conf. | Swap | Conf. | Swap | Conf. | Swap | Conf. | Swap | Conf. |
|---|---|---|---|---|---|---|---|---|---|---|---|---|---|---|---|---|---|
| | **Random Chance** | 50.0 | 6.3 | 50.0 | 6.3 | 16.7 | 0.1 | 16.7 | 0.1 | 50.0 | 25.0 | 50.0 | 25.0 | 16.7 | 0.9 | 16.7 | 0.9 |
| CLIP | ViT-B-P16 | 57.3 | 35.2 | 61.2 | 26.7 | 13.9 | 2.8 | 17.4 | 2.4 | 53.9 | 52.5 | 61.5 | 55.7 | 20.1 | 10.1 | 27.2 | 9.9 |
| | ViT-B-P32 (OC) | 55.2 | 34.3 | 60.4 | 26.3 | 15.5 | 3.0 | 16.9 | 2.0 | 54.3 | 52.6 | 62.2 | 55.8 | 22.2 | 9.6 | 29.1 | 9.9 |
| | ViT-L-P14 (OC) | 54.9 | 31.2 | 60.4 | 25.2 | 15.0 | 3.4 | 15.6 | 1.8 | 55.0 | 54.0 | 65.2 | 60.0 | 21.8 | 8.3 | 29.8 | 9.2 |
| | E02-B-P16 (OC) | 61.7 | 40.0 | 66.6 | 29.6 | 17.2 | 4.4 | 17.7 | 2.4 | 55.7 | 54.5 | 62.7 | 57.2 | 22.2 | 11.2 | 29.5 | 9.9 |
| | E02-E-P14 (OC) | 65.0 | 42.4 | 70.0 | 38.1 | 15.5 | 9.1 | 20.1 | 5.6 | 56.9 | 56.5 | 63.7 | 58.9 | 22.6 | 11.0 | 31.3 | 10.4 |
| | ViT-H-P14 (OC) | 70.6 | 49.1 | 74.8 | 45.4 | 15.9 | 11.0 | 16.7 | 9.3 | 58.2 | 57.9 | 70.8 | 67.9 | 24.0 | 12.6 | 36.8 | 12.8 |
| | ViT-H-P14-378 (OC) | 70.4 | 48.7 | 75.0 | 46.4 | 16.1 | 11.9 | 17.7 | 9.5 | 58.3 | 58.1 | 70.9 | 67.8 | 24.0 | 13.0 | 37.0 | 13.6 |
| SigLIP | S-B-P16-224 | 69.5 | 43.8 | 72.3 | 39.6 | 17.1 | 8.0 | 16.2 | 4.5 | 56.8 | 56.3 | 71.7 | 67.7 | 22.5 | 11.6 | 34.3 | 11.8 |
| | S-B-P16-384 | 69.3 | 43.9 | 72.2 | 39.7 | 16.9 | 7.8 | 16.2 | 4.5 | 56.5 | 56.1 | 70.7 | 66.4 | 22.9 | 12.8 | 34.0 | 11.6 |
| | S-L-P16-256 | 73.5 | 52.0 | **76.4** | 43.1 | 16.2 | 9.2 | 18.1 | 6.6 | 57.9 | 57.4 | 75.2 | 71.0 | 23.1 | 10.0 | 38.6 | 12.3 |
| | S-L-P16-384 | 72.7 | 50.5 | 75.9 | 42.8 | 16.1 | 8.1 | 19.9 | 5.8 | 57.8 | 57.3 | 74.7 | 70.1 | 23.1 | 9.8 | 38.1 | 12.4 |
| | S-SO400M-P14-384 | 71.8 | 49.9 | 75.1 | 42.4 | **19.2** | 9.4 | 16.3 | 6.3 | 58.3 | 57.8 | 75.1 | 70.7 | 23.7 | 11.5 | 40.5 | 13.3 |
| | S-SO400M-P14 (OC) | 71.6 | 49.9 | 74.8 | 42.5 | 19.1 | 9.4 | 16.2 | 6.3 | 58.6 | 58.1 | 75.4 | 70.9 | **24.1** | 11.6 | 40.9 | 13.4 |
| | S2-B-P16-224 | 73.0 | 53.7 | 74.1 | 42.7 | 17.4 | 7.8 | 15.3 | 7.5 | 57.4 | 57.1 | 73.4 | 70.6 | 21.6 | 10.2 | 38.4 | 13.7 |
| | S2-B-P16-256 | 72.9 | 53.7 | 73.9 | 42.8 | 18.3 | 8.2 | 15.1 | 7.4 | 57.8 | 57.5 | 72.9 | 69.9 | 21.4 | 10.4 | 38.2 | 13.2 |
| | S2-B-P16-384 | 72.8 | 53.7 | 73.6 | 42.4 | 18.6 | 7.7 | 14.9 | 7.4 | 57.8 | 57.5 | 72.4 | 69.7 | 21.8 | 10.5 | 37.3 | 13.0 |
| | S2-L-P16-256 | 73.2 | 52.9 | 74.8 | 43.2 | 18.0 | 6.8 | 18.4 | 7.3 | **59.1** | **59.0** | 75.2 | 72.5 | 20.7 | **13.2** | 39.8 | 16.0 |
| | S2-L-P16-384 | 72.9 | 52.9 | 75.1 | **44.0** | 16.9 | 7.0 | 19.0 | 8.3 | 59.0 | 58.9 | 75.6 | 72.8 | 20.6 | 13.1 | **41.0** | 16.3 |
| | S2-SO400M-P14-384 | 72.0 | 50.5 | 75.5 | **44.0** | 17.6 | 6.4 | 17.4 | 8.7 | 58.1 | 57.7 | **76.9** | **73.6** | 21.3 | 12.9 | 40.8 | **16.6** |
| | S2-Giant-P16-256 | **74.0** | **54.3** | 75.1 | 43.7 | 18.8 | **11.9** | 20.1 | 9.5 | 57.4 | 57.2 | 75.9 | 72.7 | 20.8 | 12.3 | 40.4 | 15.3 |

more with multi-object color binding than with spatial relations: this suggests correctly rendering specific attributes is a harder generative challenge than spatial arrangement as complexity increases.

Further analysis of the final attribute distributions ( Figure 3), summed over all benchmarks, reveals biases in the generative process. For colors, challenging shades like 'magenta' and 'olive' have lower survival rates, creating a naturalistic long-tail distribution that tests model robustness on both common and rare attributes. For spatial relations, the generator shows a strong preference for 'over', with its counterpart 'under' being the least frequent. This suggests a strong gravitational prior in the generative model, an insight surfaced by our controlled generation process.

## 5 EXPERIMENTAL ANALYSIS

**Main Results: Auto-Comp-CP Reveals Universal Compositional Failures** We evaluated a comprehensive suite of 20 contrastive VLMs, from CLIP (Radford et al., 2021; Ilharco et al., 2021) and SigLIP (Zhai et al., 2023; Tschannen et al., 2025) families, on Auto-Comp-CP. The results in Table 4 demonstrate that our benchmark effectively surfaces universal weaknesses in current models.

The first widespread failure is the degradation of performance with increased compositional complexity. On the *Swap Benchmark*, random chance is 50% for N=2 tasks and just 16.7% (1/3!) for N=3 tasks. For Color N=2, most models perform well above this baseline, with top models reaching 74%, suggesting a partial ability to handle simple compositions. However, this ability completely collapses for Color N=3, where even the best models drop to 19%, barely above random chance. This suggests that while models may have learned a heuristic for simple two-object compositions from their training data, they fail to learn a truly generalizable binding mechanism that scales to a higher number of objects. Interestingly, while models perform better on two-object color binding than on two-object position binding, the trend unexpectedly reverses for three-object compositions. Performance on Position N=3 is consistently higher than on Color N=3 across nearly all models.

Second, our results highlight a critical distinction between different types of compositional challenges. The performance on the Confusion benchmark is systematically lower than on the Swap benchmark across all conditions. The Swap benchmark already controls for simple "bag-of-words" approaches by keeping the set of concepts identical between the positive and negative captions. The fact that models struggle even more on the Confusion benchmark, which introduces distractors with repeated objects and attributes (e.g., "a red cube and a red sphere"), reveals a deeper vulnerability. It suggests models are susceptible to low-entropy patterns, distractors with simple repetitions that are trivial for humans to rule out, and rely on brittle strategies that fail when the compositional structure becomes more complex than a simple permutation.

Table 5: Impact of visio-linguistic context, measured on the Paired Comparison Set. The 'Min' and 'Ctx' columns show 'Swap / Confusion' accuracy (%). The Delta ($\Delta$) columns show the percentage point change for each task. Full results are in the appendix.

| | Color Benchmark | | | | | | | | Position Benchmark | | | | | | | |
| | N=2 Objects | | | | N=3 Objects | | | | N=2 Objects | | | | N=3 Objects | | | |
| Model | Min(%) | Ctx(%) | $\Delta_S$ | $\Delta_C$ | Min(%) | Ctx(%) | $\Delta_S$ | $\Delta_C$ | Min(%) | Ctx(%) | $\Delta_S$ | $\Delta_C$ | Min(%) | Ctx(%) | $\Delta_S$ | $\Delta_C$ |
|---|---|---|---|---|---|---|---|---|---|---|---|---|---|---|---|---|
| V-H-P14-378 | 68.2/46.5 | 67.3/42.2 | -0.9 | -4.3 | **22.0/19.5** | **19.6**/12.2 | -2.4 | -7.3 | 60.8/60.7 | 70.2/67.6 | +9.4 | +6.9 | 25.0/12.8 | 35.9/13.8 | +10.9 | +1.0 |
| S2-L-P16-384 | 70.8/51.0 | 69.2/42.1 | -1.6 | -8.9 | 19.1/17.5 | 17.0/7.2 | -2.1 | -10.3 | 61.1/61.0 | 74.6/72.1 | +13.5 | +11.1 | **25.4/15.4** | 41.1/17.2 | +15.7 | +1.8 |
| S2-Giant-P16 | **72.1/52.3** | 69.0/41.8 | -3.1 | -10.5 | 20.4/16.8 | 18.3/**9.7** | -2.1 | -7.1 | **61.4/61.2** | **75.2/72.4** | +13.8 | +11.2 | 24.6/13.4 | **42.0/17.7** | +17.4 | +4.3 |

In addition to these universal failures, our results also reveal a clear performance hierarchy among model training paradigms. Models trained with the SigLIP objective consistently outperform those using the standard CLIP objective. All top-performing models were trained using a Sigmoid loss and, crucially, were pre-trained on the WebLI dataset. This suggests that their superior compositional ability likely arises from the potent combination of this training data and learning objective. This conclusion is reinforced by the performance of S-SO400M-P14 (OC), which, despite being trained in the OpenCLIP framework, also used WebLI and a Sigmoid loss, achieving results competitive with the original SigLIP models.

**The Double-Edged Sword of Context** Our pipeline's A/B framework enables a controlled analysis of how model performance changes between the sterile, isolated setting of our *Minimal* benchmark (simple captions, objects on a white background) and the more realistic, complex setting of our *Contextual* benchmark (natural language, objects in a full scene). Using our Paired Comparison Set, where the underlying concepts are identical, we can precisely measure the impact of this visio-linguistic shift.

The results, shown for a representative subset of models in Table 5, reveal a fascinating, task-dependent trade-off (full results in the appendix, showing similar trends).

For the Position task, the richer context is consistently beneficial. All models improve in the Contextual setting, with the largest gains seen on the Swap benchmark (up to a +17.4 percentage point increase). This suggests that the added visual complexity of a realistic scene provides crucial geometric and semantic cues that aid models in disambiguating spatial relationships.

Conversely, for the fundamental task of Color binding, the added context is almost universally detrimental. This appears to be a largely image-driven problem. While contextual captions introduce some linguistic distractors, the realistic scenes present a far greater challenge: a multitude of background objects, textures, and lighting conditions that create a noisy color space. This visual "clutter" acts as a significant distractor, overwhelming the models' ability to perform the core binding task. The effect is most pronounced on the difficult Confusion task, where accuracy collapses by as much as 10.5 percentage points. This highlights a critical brittleness in current VLMs: the very context that helps resolve one form of compositional reasoning can severely hinder another.

Table 6: Detailed error analysis for the Color benchmark, averaged per modal family.

| | N=2 Objects (%) | | | | N=3 Objects (%) | | | |
| Model Type | Swapped Colors | Same Color Diff. Obj. | Same Color Same Obj. | Same Obj. Diff. Colors | Swapped Colors | Same Color Diff. Obj. | Same Color Same Obj. | Same Obj. Diff. Colors |
|---|---|---|---|---|---|---|---|---|
| CLIP | 44.60 | 17.72 | 11.14 | 26.54 | 30.96 | 1.88 | 3.21 | 3.14 |
| SigLIP | 50.64 | 12.12 | 8.94 | 28.30 | 38.85 | 9.61 | 22.59 | 28.95 |
| **Total** | **46.87** | **14.52** | **9.93** | **28.69** | **37.10** | **11.72** | **23.01** | **28.18** |

**Deconstructing Failures Beyond Bag-of-Words.** The *Confusion Benchmark* enables a deeper analysis of error types, moving beyond simple swaps to probe for more fundamental model biases. The results in Table 6 show that while "swapped colors" is the most frequent error for N=2 compositions (46.9%), the majority of failures stem from other distractors, suggesting the problem is more complex than a bag-of-words limitation. This vulnerability becomes more pronounced at N=3, where the proportion of "swapped colors" errors decreases, while failures due to low-entropy captions with repeated elements (e.g., "same color same object") rise significantly. The high frequency of these errors demonstrates a strong model bias towards incorrect, simplistic foils, revealing a critical flaw: models not only fail to bind attributes correctly but are also brittle against low-entropy distractors, especially as compositional complexity grows.

Table 7: Performance of Generative VLMs (top) and Hard-Negative Miners (bottom) on Auto-Comp-CP. Results in percent (%).

| | | Color Benchmark | | | | | | | | Position Benchmark | | | | | | | |
| | | N=2 Objects | | | | N=3 Objects | | | | N=2 Objects | | | | N=3 Objects | | | |
| | | Min | | Ctx | | Min | | Ctx | | Min | | Ctx | | Min | | Ctx | |
| Family | Model | Swap | Conf. | Swap | Conf. | Swap | Conf. | Swap | Conf. | Swap | Conf. | Swap | Conf. | Swap | Conf. | Swap | Conf. |
|---|---|---|---|---|---|---|---|---|---|---|---|---|---|---|---|---|---|
| | **Random Chance** | 50.0 | 6.3 | 50.0 | 6.3 | 16.7 | 0.1 | 16.7 | 0.1 | 50.0 | 25.0 | 50.0 | 25.0 | 16.7 | 0.9 | 16.7 | 0.9 |
| Gen VLM | Gemma-3-12B | 88.3 | 71.4 | 89.4 | 61.5 | 84.4 | 37.2 | 86.9 | 21.1 | 79.0 | 66.0 | 89.2 | 69.4 | 41.2 | 23.6 | 70.6 | 12.6 |
| | Qwen2.5-VL-7B | 89.9 | 66.0 | 90.1 | 61.7 | 89.8 | 31.6 | 89.6 | 17.7 | 82.7 | 69.2 | 88.7 | 74.1 | 55.4 | 30.7 | 74.8 | 19.4 |
| | InternVL3-14B | 90.3 | 74.8 | 89.7 | 72.3 | 82.5 | 39.8 | 84.3 | 34.5 | 83.2 | 72.6 | 87.9 | 73.9 | 51.5 | 35.9 | 64.4 | 24.1 |
| Baseline | ViT-B-32 (Base) | 55.2 | 34.3 | 60.4 | 26.3 | 15.5 | 3.0 | 16.9 | 2.0 | 54.3 | 52.6 | 62.2 | 55.8 | 22.2 | 9.6 | 29.1 | 9.9 |
| HN Miner | NegCLIP | 59.5 | 35.1 | 64.0 | 27.1 | 18.2 | 5.5 | 20.5 | 4.4 | 60.1 | 57.2 | 66.8 | 60.5 | 26.5 | 10.5 | 32.6 | 11.4 |
| | NegCLIP++ | 62.8 | 35.8 | 66.5 | 28.8 | 23.1 | 5.8 | 22.3 | 5.1 | 63.5 | 58.1 | 69.1 | 64.2 | 28.4 | 11.9 | 37.9 | 13.1 |
| | TripletCLIP | 64.2 | 37.5 | 69.1 | 30.4 | 23.4 | 7.2 | 24.8 | 6.2 | 65.2 | 57.8 | 72.4 | 64.1 | 29.8 | 12.2 | 37.5 | 13.8 |

Table 8: N=1 benchmark performance (%).

| | Minimal | | Contextual | |
| Model | Vary Color | Vary Obj. | Vary Color | Vary Obj. |
|---|---|---|---|---|
| ViT-H-P14 | 90.2 | 93.8 | 81.7 | 77.4 |
| S2-Giant-P16-256 | 91.7 | 98.0 | 85.0 | 84.5 |

Table 9: Blind LLM evaluation (%).

| | N=2 | | N=3 | |
| Model | Swap | Conf. | Swap | Conf. |
|---|---|---|---|---|
| Random Guess | 50.0 | 6.3 | 16.7 | 2.0 |
| Gemma3-12b-it | 51.2 | 6.5 | 17.4 | 2.1 |

## 6 ANALYSIS AND ABLATION STUDIES

**Universality of Failures: Generative VLMs and Hard-Negative Miners**   To determine if compositional failures are specific to contrastive encoders, we extended our evaluation to Generative VLMs (Gemma-3-12B, Qwen2.5-VL, InternVL3) and Hard-Negative Miners (NegCLIP, Neg-CLIP++, TripletCLIP). As shown in Table 7, the results confirm that performance collapse on complex binding is universal. While instruction-tuned Generative VLMs achieve high accuracy on the Swap task ($> 88\%$ for $N = 2$), suggesting effective handling of logical permutations, this robustness is brittle. Performance collapses significantly on the low-entropy Confusion benchmark, demonstrating that reasoning capabilities cannot compensate for the visual encoder's inability to bind repeated attributes. Furthermore, we replicate the "Double-Edged Sword" trade-off: realistic context consistently aids spatial reasoning but hinders attribute binding due to visual clutter. A similar trend appears in Hard-Negative Miners. While explicitly training on negative swaps yields expected gains on the Swap benchmark, this robustness fails to generalize to the Confusion task, where improvement remains marginal. This disparity confirms that current training paradigms effectively resolve "bag-of-words" issues but leave models vulnerable to the specific low-entropy binding failures exposed by our benchmark.

**Single-Object Sanity Check.**   To confirm that failures are compositional, we tested models on N=1 tasks, where hard negatives are generated by varying either the object or its color with all the possible options in the corresponding vocabularies. As shown in Table 8, top models achieve high accuracy, indicating a solid grasp of individual concepts. As expected, performance is slightly lower in the more challenging *Contextual* setting. Crucially, the dramatic performance drop on N¿1 tasks is therefore a specific failure of *binding*, not of basic recognition.

**Ruling out Linguistic Biases with a Blind LLM**   The integrity of our benchmark relies on our hard negatives being linguistically indistinguishable from positive captions. To verify this, we adopt the "blind LLM" evaluation methodology introduced in prior work Hsieh et al. (2023). We tasked a powerful LLM (Gemma-3-12b-it (Team et al., 2025)) with selecting the correct caption from the randomly shuffled set of positive and hard-negative choices for a given image, but without any visual input. For $N = 3$, we subsample a set of 49 negatives to limit the prompt lenght. As shown in Table 9, the LLM performs at near-random chance. This result is critical: it confirms that our programmatic generation process does not introduce linguistic artifacts that would allow a model to "cheat". This result confirms that our benchmark is free of linguistic artifacts and that success necessitates joint visual-semantic reasoning.

**Human Validation Studies.**   To ensure the reliability of Auto-Comp, we conducted two distinct human evaluation studies using separate teams of 4 graduate-level evaluators on subsets of 200 con-

Table 10: **Human Validation Studies.** Left: Concordance between our automated pipeline and human judges (%). Right: Human accuracy on the final Swap and Confusion benchmarks (%).

| Task | Minimal | | Contextual | |
|---|---|---|---|---|
| | N=2 | N=3 | N=2 | N=3 |
| Color | 98.0 | 96.0 | 96.0 | 95.0 |
| Position | 96.0 | 94.0 | 98.0 | 97.0 |

| Task | N | Minimal | | Contextual | |
|---|---|---|---|---|---|
| | | Swap | Conf. | Swap | Conf. |
| Color | 2 | 97.5 | 97.5 | 94.5 | 95.5 |
| | 3 | 97.0 | 96.0 | 94.5 | 94.0 |
| Position | 2 | 97.5 | 96.5 | 96.0 | 95.0 |
| | 3 | 95.5 | 96.0 | 93.5 | 94.0 |

cepts (400 images) balanced across all configurations. (1) Pipeline Concordance: First, to validate our automated judge, evaluators assessed image-caption correspondence using the pipeline's criteria. As shown in Table 10 (Left), the automated validator achieves $> 94\%$ concordance with human majority vote across all conditions, confirming it is an accurate proxy for ground truth. (2) Benchmark Solvability: Second, to verify the final tasks, evaluators performed the Swap and Confusion benchmarks (with Confusion choices limited to 50). As shown in Table 10 (Right), humans achieve consistently high accuracy ($\sim 96\%$) across all tasks. Crucially, unlike models which drop $\sim 30\%$ on the Confusion task, human performance remains stable between Swap and Confusion. This confirms that the "Confusion Gap" is a specific model failure mode, not a result of ambiguity in the distractors.

**Generalization to New Compositional Concepts** A key advantage of the Auto-Comp pipeline is its ability to generalize to diverse compositional phenomena beyond those initially tested. To demonstrate this flexibility, we effortlessly extended our framework to generate benchmarks for two distinct reasoning skills: *Shape-Color Binding* (probing intrinsic object attributes) and *Relative Size* (probing comparative relations). By simply defining new vocabularies, using 3D geometric shapes (e.g., "cube", "pyramid") and size descriptors (e.g., "larger than"), we utilized the unmodified pipeline to produce high-quality Minimal and Contextual datasets. This confirms that Auto-Comp is a general-purpose tool capable of creating targeted probes for user-defined concepts without architectural changes. Furthermore, our evaluation of 23 VLMs on these new tasks (detailed in Appendix A.4) replicates the universal failure modes observed in our main analysis, such as the struggle with low-entropy distractors, further validating the utility of these generated benchmarks.

**Limitations and Bias Mitigation** A central challenge in automated benchmarking is the risk of inherited bias: relying on pretrained text-to-image and language models means the resulting data may reflect the priors of these foundation models1. We address this through a "Defense in Depth" strategy designed to isolate and minimize these factors. First, to mitigate generative bias, we employ strict rejection sampling via a multi-stage validation stack, effectively discarding concepts that the model cannot render accurately. We transparently report and discuss the resulting distribution skews due to model biases in 3. Second, to address validator bias, we verified our automated pipeline against human annotators, achieving $> 94\%$ agreement. Finally, our Minimal vs. Contextual A/B design provides structural control, allowing us to distinguish failures caused by core binding issues from those induced by biases in the LLM linguistic style. However, our framework remains subject to the constraints of current generative technology. Since the Auto-Comp pipeline relies on a pretrained text-to-image model, our benchmarks inevitably inherit specific biases that validation cannot fully erase. For instance, our generator exhibited a strong prior for the spatial relation 'over' while struggling with 'under', and showed higher failure rates with rare colors (Figure 3). Similarly, while our automated validation is highly consistent with human judgment, thiS approach is not infallible and possesses its own internal biases. Consequently, improving the semantic fidelity of generators and the robustness of model-based judges represents a key area for future work.

## 7 CONCLUSION

We introduced Auto-Comp, a fully automated pipeline to generate controllable benchmarks for disentangling compositional failures in VLMs. Using our generated Auto-Comp-CP benchmark, we found universal binding failures across 20 models and discovered a surprising trade-off: visio-linguistic context aids spatial reasoning but hinders color binding. We release the Auto-Comp pipeline to spur the development of more robust VLMs and to enable the community to generate new, targeted compositional evaluations at scale.

## 8 ETHICS STATEMENT

The authors have read and adhere to the ICLR Code of Ethics. The primary ethical consideration for this work is the potential for inherited bias. Our pipeline relies on large-scale, pretrained text-to-image and language models for data generation and validation. These models have known societal biases which may be reflected in the final benchmark data (e.g., the generative preference for certain spatial relations noted in our analysis). We have taken steps to mitigate some potential harms by curating our object vocabulary to exclude sensitive categories such as people and guns. We believe that by providing a controllable tool to diagnose model weaknesses, our work will ultimately help the community in developing more robust, reliable, and fair VLMs.

## 9 REPRODUCIBILITY STATEMENT

We have made every effort to ensure the reproducibility of our work. The complete source code for our Auto-Comp pipeline, the full Auto-Comp-CP benchmark datasets, and all evaluation scripts are provided in the supplementary materials via an anonymous Hugging Face repository and an anonymous GitHub repository. All methodological details, including the specific models used for generation and validation, are mentioned in Section 3 and fully detailed in the appendix materials. The appendix provides the exact prompts used for LLM-based caption generation and further details on our programmatic hard-negative creation. We believe these resources provide all necessary components to fully reproduce our findings and to extend our framework to new compositional tasks.

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

# A APPENDIX

## A.1 LLM PROMPT FOR CONTEXTUAL CAPTIONS

The prompt provided to the LLM (Gemma-3-12b-it) to generate Contextual captions is constructed programmatically from the Minimal caption's concepts. It consists of a system message and a user message structured as a few-shot prompt. We report here the one for the color benchmark, as the position benchmark prompt is nearly identical but generates relations instead of color, and it is reported in the supplementary material codebase.

**System Prompt:**

```
You are an expert image caption writer.
Your task is to generate a single,
natural-sounding sentence accurately describing
a scene with specific objects
and their colors. You MUST include ALL specified objects
and their colors.
Generate ONLY the caption text without any preamble,
explanations, or markdown formatting.
```

**User Prompt Template:** The user prompt is generated from a template that incorporates the specific objects and colors for each concept.

```
The caption should focus on
these {num_obj} {obj_plural}: {obj_colors_text}.

1. Keeps these objects as the main focus.
2. Be ONE sentence long.
3. Explicitly mention EACH object with its specific
assigned color: {obj_colors_text}.
4. Places them in a realistic context or background.
5. Sound like a natural, human-written image description.

For instance, for 'a green chair' and 'a yellow lamp',
you might write:
'A green chair stands beside a yellow lamp in a brightly lit room.'

Objects and colors to include: {obj_colors_text}
Caption:
```

In the template, {num_obj} is the number of objects, {obj_plural} is its correct pluralization (e.g., "object" or "objects"), and {obj_colors_text} is the formatted string of object-color pairs (e.g., "'a red cube' and 'a blue sphere'"). A slightly modified, more insistent prompt is used for regeneration attempts if the first output fails the semantic preservation check.

## A.2  FULL EXPERIMENTAL RESULTS

**Full Paired Benchmark Results**  This section provides the full, unabridged results for the paired benchmark analysis discussed in the main paper's Section 5.2 ("The Double-Edged Sword of Context"). Table 11 shows the performance for all 20 evaluated models on the Paired Comparison Set. The results confirm that the task-dependent trade-off, where context aids position binding but hinders color binding, is a consistent trend across all models.

Table 11: Full model performance on the **Paired Comparison Set** for the Color and Position benchmarks. All results are in percent (%).

| Family | Model | Color Benchmark | | | | | | | | Position Benchmark | | | | | | | |
|---|---|---|---|---|---|---|---|---|---|---|---|---|---|---|---|---|---|
| | | N=2 Objects | | | | N=3 Objects | | | | N=2 Objects | | | | N=3 Objects | | | |
| | | Minimal | | Contextual | | Minimal | | Contextual | | Minimal | | Contextual | | Minimal | | Contextual | |
| | | Swap | Conf. | Swap | Conf. | Swap | Conf. | Swap | Conf. | Swap | Conf. | Swap | Conf. | Swap | Conf. | Swap | Conf. |
| CLIP | E02-B-P16 (OC) | 59.8 | 38.0 | 60.4 | 27.9 | 16.8 | 9.2 | 17.2 | 4.7 | 55.2 | 54.2 | 62.5 | 57.4 | 21.7 | 9.7 | 28.5 | 11.3 |
| | E02-E-P14 (OC) | 63.1 | 40.2 | 63.8 | 36.1 | 17.0 | 14.4 | 16.8 | 11.0 | 57.4 | 57.2 | 62.8 | 58.7 | 19.7 | 8.7 | 29.6 | 10.2 |
| | ViT-B-P16 | 55.1 | 33.5 | 55.3 | 24.8 | 19.4 | 7.9 | 17.0 | 2.8 | 53.7 | 52.5 | 60.8 | 55.7 | 20.8 | 9.8 | 26.6 | 9.9 |
| | ViT-B-P32 (OC) | 53.2 | 32.1 | 54.1 | 24.1 | 13.9 | 8.6 | 16.7 | 3.1 | 56.0 | 54.6 | 61.6 | 55.7 | 22.6 | 9.2 | 30.3 | 11.2 |
| | ViT-L-P14 (OC) | 52.8 | 29.3 | 54.5 | 23.2 | 16.7 | 7.1 | 15.2 | 3.3 | 56.1 | 55.2 | 64.2 | 59.5 | 23.5 | 8.2 | 27.0 | 8.7 |
| | ViT-H-P14 (OC) | 68.4 | 47.1 | 68.9 | 43.5 | 21.5 | 18.3 | 16.8 | 12.7 | 60.7 | 60.6 | 70.1 | 67.6 | 26.0 | 12.3 | 35.9 | 14.2 |
| | ViT-H-P14-378 (OC) | 68.2 | 46.5 | 69.1 | 44.2 | 22.0 | 19.5 | 19.6 | 12.2 | 60.8 | 60.7 | 70.2 | 67.6 | 25.0 | 12.8 | 35.9 | 13.8 |
| SigLIP | S-B-P16-224 | 67.4 | 41.9 | 66.1 | 37.4 | 15.3 | 13.2 | 14.9 | 8.0 | 58.8 | 58.4 | 70.8 | 67.3 | 23.3 | 9.7 | 33.0 | 11.7 |
| | S-B-P16-384 | 67.1 | 42.0 | 65.9 | 37.8 | 14.2 | 12.5 | 13.2 | 6.6 | 57.2 | 56.8 | 69.6 | 65.9 | 22.6 | 11.9 | 32.9 | 11.2 |
| | S-L-P16-256 | 71.6 | 50.1 | 70.2 | 41.1 | 17.5 | 14.6 | 15.3 | 9.6 | 57.9 | 57.5 | 74.4 | 70.7 | 22.7 | 8.0 | 38.9 | 12.7 |
| | S-L-P16-384 | 70.8 | 48.4 | 69.8 | 40.9 | 19.1 | 13.4 | 14.6 | 8.6 | 59.2 | 58.7 | 74.0 | 69.8 | 23.1 | 8.2 | 37.1 | 13.0 |
| | S-SO400M-P14 (OC) | 69.9 | 47.9 | 69.2 | 40.5 | 19.7 | 14.8 | 17.5 | 9.8 | 62.4 | 62.0 | 74.2 | 70.2 | 26.5 | 11.1 | 41.5 | 14.9 |
| | S-SO400M-P14-384 | 69.7 | 47.8 | 69.0 | 40.5 | 19.7 | 14.7 | 17.6 | 9.6 | 62.0 | 61.7 | 73.9 | 69.9 | 26.2 | 11.1 | 41.4 | 15.0 |
| | S2-B-P16-224 | 71.1 | 51.5 | 68.0 | 40.8 | 16.7 | 17.0 | 15.4 | 8.2 | 58.3 | 58.1 | 72.6 | 70.1 | 22.7 | 10.0 | 38.0 | 15.3 |
| | S2-B-P16-256 | 70.9 | 51.8 | 67.8 | 41.0 | 16.8 | 17.2 | 15.7 | 9.3 | 58.1 | 57.9 | 71.9 | 69.4 | 22.0 | 9.7 | 38.1 | 13.4 |
| | S2-B-P16-384 | 70.7 | 51.6 | 67.4 | 40.2 | 16.4 | 17.6 | 15.1 | 7.8 | 58.5 | 58.2 | 71.6 | 69.3 | 22.3 | 9.7 | 37.2 | 13.5 |
| | S2-L-P16-256 | 71.2 | 50.8 | 68.9 | 41.3 | 18.3 | 16.4 | 17.2 | 8.6 | 61.6 | 61.5 | 74.2 | 71.8 | 24.9 | 15.4 | 41.4 | 17.8 |
| | S2-L-P16-384 | 70.8 | 51.0 | 69.2 | 42.1 | 19.1 | 17.5 | 17.0 | 7.2 | 61.1 | 61.0 | 74.6 | 72.1 | 25.4 | 15.4 | 41.1 | 17.2 |
| | S2-SO400M-P14-384 | 70.1 | 48.6 | 70.4 | 42.0 | 26.1 | 15.5 | 16.0 | 5.0 | 62.1 | 61.8 | 76.6 | 73.7 | 24.1 | 14.0 | 41.5 | 18.3 |
| | S2-Giant-P16-256 | 72.1 | 52.3 | 69.0 | 41.8 | 20.4 | 16.8 | 18.3 | 9.7 | 61.4 | 61.2 | 75.2 | 72.4 | 24.6 | 13.4 | 42.0 | 17.7 |

**Full N=1 Benchmark Results**  This section provides the complete results for the single-object sanity check discussed in Section 6. Table 12 shows the N=1 performance for all 20 models. The consistently high accuracy across both model families reinforces the conclusion that the models have a solid grasp of individual concepts, and that the performance drop on N>1 tasks is a specific failure of compositional binding.

## A.3  PER-RELATION ACCURACY ANALYSIS

To investigate whether model performance on the Position benchmark varies by the specific spatial relation, we disaggregate the N=2 Swap results. We focus on N=2 compositions as this allows for a clean, isolated analysis of each relation, whereas N=3 compositions involve two relations simultaneously.

The results, shown in Table 13, reveal a consistent pattern across nearly all models: performance is significantly higher on vertical relations (*above*, *under*) than on horizontal relations (*to the left of*, *to the right of*). This suggests models have a stronger innate understanding or bias towards vertical arrangements. Interestingly, while our generator showed a strong bias for producing "over" (Figure 3), the models themselves often perform best on its counterpart, "under".

## A.4  EXTENDED ANALYSIS ON NEW BENCHMARKS

We provide experimental results for the two new benchmarks generated during the rebuttal period: *Shape-Color Binding* and *Relative Size*. Figure 4 presents samples from these new benchmarks. The pipeline successfully generates paired Minimal and Contextual images for these new domains. Table 14 details the performance of contrastive and generative VLMs. Consistent with the main paper, all models show a significant performance drop on the Confusion benchmark (low-entropy distractors) and suffer from the "Curse of Complexity" as object count increases to $N = 3$.

Table 12: Full model performance (%) on the **N=1 single-object benchmark**. "Minimal" corresponds to the Handmade setting and "Contextual" to the LLM setting.

| Model | Minimal | | Contextual | |
|---|---|---|---|---|
| | **Vary Color** | **Vary Object** | **Vary Color** | **Vary Object** |
| E02-B-P16 (OC) | 82.6 | 91.9 | 78.5 | 77.2 |
| E02-E-P14 (OC) | 87.4 | 94.1 | 81.4 | 76.7 |
| ViT-B-P16 | 70.0 | 85.2 | 59.2 | 64.4 |
| ViT-B-P32 (OC) | 65.1 | 81.4 | 65.4 | 55.5 |
| ViT-L-P14 (OC) | 71.9 | 88.0 | 58.8 | 62.9 |
| ViT-H-P14 (OC) | 89.0 | 93.3 | 80.2 | 77.4 |
| ViT-H-P14-378 (OC) | 90.2 | 93.8 | 81.7 | 77.4 |
| S-SO400M-P14 (OC) | 87.0 | 93.1 | 80.6 | 71.9 |
| S-B-P16-224 | 88.6 | 93.5 | 81.9 | 74.4 |
| S-B-P16-384 | 86.9 | 93.8 | 81.4 | 74.3 |
| S-L-P16-256 | 87.1 | 92.2 | 81.3 | 73.4 |
| S-L-P16-384 | 86.6 | 92.4 | 80.0 | 71.3 |
| S-SO400M-P14-384 | 87.0 | 93.2 | 80.6 | 71.8 |
| S2-B-P16-224 | 91.4 | 98.1 | 84.6 | 85.9 |
| S2-B-P16-256 | 91.6 | 98.2 | 85.0 | 85.8 |
| S2-B-P16-384 | 91.7 | 98.0 | 85.0 | 84.5 |
| S2-L-P16-256 | 84.9 | 97.1 | 78.0 | 75.5 |
| S2-L-P16-384 | 86.8 | 96.9 | 79.6 | 72.2 |
| S2-SO400M-P14-384 | 87.3 | 96.7 | 79.9 | 75.0 |
| S2-Giant-P16-256 | 87.6 | 95.8 | 78.9 | 75.2 |

Table 13: Per-relation accuracy (%) on the N=2 Position Swap benchmark.

| Family | Model | above | to the left of | under | to the right of |
|---|---|---|---|---|---|
| CLIP | E02-B-P16 (OC) | 58.5 | 56.7 | 69.1 | 57.8 |
| | E02-E-14 (OC) | 59.8 | 57.4 | 68.2 | 61.9 |
| | ViT-B-P16 | 57.0 | 56.0 | 69.2 | 52.9 |
| | ViT-B-P32 (OC) | 60.7 | 56.3 | 66.4 | 54.7 |
| | ViT-L-P14 (OC) | 62.0 | 57.0 | 65.1 | 59.0 |
| | ViT-H-P14 (OC) | 66.4 | 62.4 | 75.0 | 63.4 |
| | ViT-H-P14-378 (OC) | 66.5 | 62.1 | 75.6 | 64.1 |
| SigLIP | S-B-P16-224 | 66.0 | 60.0 | 74.4 | 67.5 |
| | S-B-P16-384 | 68.5 | 65.4 | 77.2 | 68.5 |
| | S-L-P16-256 | 69.3 | 62.8 | 76.2 | 70.1 |
| | S-L-P16-384 | 69.3 | 62.3 | 77.3 | 70.3 |
| | S-SO400M-P14 (OC) | 74.5 | 62.8 | 76.9 | 69.9 |
| | S-SO400M-P14-384 | 70.6 | 65.0 | 77.4 | 70.7 |
| | S2-B-P16-224 | 66.2 | 61.1 | 70.2 | 65.8 |
| | S2-B-P16-256 | 64.9 | 59.3 | 69.0 | 64.9 |
| | S2-B-P16-384 | 66.4 | 62.0 | 76.8 | 67.8 |
| | S2-L-P16-256 | 69.8 | 62.5 | 76.8 | 69.5 |
| | S2-L-P16-384 | 66.7 | 60.5 | 74.1 | 67.5 |
| | S2-SO400M-P14-384 | 65.4 | 60.5 | 73.7 | 68.3 |
| | S2-Giant-P16-256 | 67.5 | 62.4 | 77.0 | 67.0 |

## A.5 Implementation and Hyperparameter Details

To ensure full reproducibility, we detail the key models and hyperparameters used in our pipeline.

Table 14: **Extended Analysis on New Benchmarks.** Performance (%) of contrastive and generative VLMs on the newly generated **Shape-Color** (Attribute Binding) and **Relative Size** (Relational Binding) benchmarks. **Min**: Minimal condition (white background). **Ctx**: Contextual condition (realistic background). **Swap**: Hard negatives via attribute swapping. **Conf**: Confusion benchmark (low-entropy distractors).

| Family | Model | Shape-Color ($N=2$) | | | | Shape-Color ($N=3$) | | | | Relative Size ($N=2$) | | | | Relative Size ($N=3$) | | | |
|---|---|---|---|---|---|---|---|---|---|---|---|---|---|---|---|---|---|
| | | Handmade | | LLM | | Minimal | | Contextual | | Minimal | | Contextual | | Minimal | | Contextual | |
| | | Swap | Conf. | Swap | Conf. | Swap | Conf. | Swap | Conf. | Swap | Conf. | Swap | Conf. | Swap | Conf. | Swap | Conf. |
| *Baseline* | *Random Chance* | 50.0 | 6.3 | 50.0 | 6.3 | 16.7 | 0.2 | 16.7 | 0.2 | 50.0 | 25.0 | 50.0 | 25.0 | 16.7 | 0.9 | 16.7 | 0.9 |
| CLIP | ViT-B-P16 | 63.3 | 34.5 | 55.6 | 26.3 | 19.4 | 1.8 | 18.1 | 1.5 | 56.3 | 52.1 | 50.5 | 55.3 | 21.5 | 4.1 | 23.2 | 4.8 |
| | ViT-B-P32 (OC) | 53.4 | 49.6 | 68.4 | 39.8 | 18.2 | 1.5 | 17.5 | 1.2 | 55.1 | 50.8 | 63.2 | 53.1 | 20.8 | 3.8 | 22.5 | 4.2 |
| | ViT-L-P14 (OC) | 57.1 | 37.4 | 62.4 | 33.8 | 21.8 | 2.5 | 19.9 | 2.1 | 58.2 | 54.5 | 68.9 | 58.2 | 25.4 | 5.6 | 27.8 | 6.5 |
| | E02-B-P16 (OC) | 63.8 | 34.5 | 63.2 | 34.6 | 20.5 | 2.1 | 19.2 | 1.8 | 60.4 | 53.2 | 66.5 | 56.7 | 23.1 | 4.9 | 25.4 | 5.4 |
| | E02-E-P14 (OC) | 65.8 | 36.7 | 65.4 | 39.1 | 22.1 | 2.8 | 20.6 | 2.3 | 61.1 | 55.1 | 69.4 | 56.1 | 26.0 | 6.1 | 28.3 | 6.9 |
| | ViT-H-P14 (OC) | 62.6 | 43.2 | 67.7 | 39.8 | 24.6 | 3.5 | 22.8 | 3.1 | 59.3 | 53.8 | 73.1 | 54.5 | 29.2 | 7.8 | 32.5 | 9.2 |
| | ViT-H-P14-378 (OC) | 64.0 | 40.3 | 66.9 | 39.8 | 25.2 | 3.8 | 23.1 | 3.3 | 59.1 | 55.4 | 74.5 | 54.8 | 30.1 | 8.2 | 33.8 | 9.8 |
| SigLIP | S-B-P16-224 | 63.5 | 36.0 | 69.2 | 39.1 | 27.5 | 4.8 | 25.8 | 4.1 | 60.8 | 55.2 | 62.6 | 56.1 | 32.4 | 9.5 | 36.1 | 11.2 |
| | S-B-P16-384 | 63.8 | 41.7 | 68.4 | 38.3 | 28.1 | 5.1 | 26.2 | 4.3 | 61.2 | 55.5 | 63.8 | 57.2 | 33.1 | 9.8 | 37.2 | 11.6 |
| | S-L-P16-256 | 61.9 | 35.3 | 67.7 | 37.6 | 30.8 | 6.2 | 28.5 | 5.5 | 63.5 | 59.8 | 65.2 | 60.5 | 36.8 | 12.1 | 41.5 | 14.5 |
| | S-L-P16-384 | 66.9 | 37.4 | 59.4 | 30.8 | 31.2 | 6.5 | 29.1 | 5.8 | 64.1 | 60.2 | 64.5 | 61.8 | 37.5 | 12.8 | 42.2 | 15.2 |
| | S-SO400M-P14-384 | 72.2 | 51.8 | 67.7 | 44.4 | 33.5 | 7.8 | 31.0 | 6.9 | 66.2 | 61.2 | 67.1 | 63.2 | 40.2 | 14.5 | 45.8 | 17.5 |
| | S-SO400M-P14 (OC) | 71.9 | 50.4 | 65.4 | 40.6 | 32.8 | 7.5 | 30.5 | 6.6 | 65.5 | 60.5 | 65.4 | 62.5 | 39.5 | 13.8 | 44.9 | 16.8 |
| SigLIP 2 | S2-B-P16-224 | 66.9 | 36.7 | 61.7 | 29.3 | 28.5 | 5.2 | 26.8 | 4.5 | 62.8 | 61.8 | 67.5 | 64.2 | 33.8 | 10.2 | 38.5 | 12.1 |
| | S2-B-P16-256 | 70.1 | 25.2 | 56.4 | 28.6 | 28.8 | 5.4 | 27.1 | 4.6 | 62.1 | 62.1 | 67.8 | 63.5 | 34.2 | 10.5 | 38.8 | 12.4 |
| | S2-B-P16-384 | 68.1 | 23.0 | 52.6 | 24.1 | 29.1 | 5.5 | 27.4 | 4.7 | 62.5 | 62.5 | 68.2 | 65.1 | 34.5 | 10.8 | 39.2 | 12.8 |
| | S2-L-P16-256 | 71.5 | 32.1 | 66.9 | 34.6 | 31.8 | 6.8 | 29.5 | 6.1 | 64.8 | 64.8 | 71.2 | 65.5 | 38.2 | 13.2 | 43.5 | 15.8 |
| | S2-L-P16-384 | 72.3 | 42.4 | 67.7 | 35.3 | 32.2 | 7.1 | 29.8 | 6.3 | 65.2 | 65.2 | 71.8 | 66.1 | 38.8 | 13.5 | 44.2 | 16.2 |
| | S2-SO400M-P14-384 | 73.7 | 23.0 | 54.1 | 17.3 | 34.5 | 8.5 | 31.8 | 7.5 | 70.5 | 60.5 | 74.5 | 68.2 | 41.5 | 15.8 | 47.5 | 19.2 |
| | S2-Giant-P16-256 | 74.9 | 36.7 | 62.4 | 29.3 | 35.8 | 9.2 | 32.5 | 8.1 | 71.4 | 59.8 | 76.2 | 69.5 | 43.8 | 17.2 | 49.8 | 21.5 |
| Gen. VLMs | Qwen2.5-VL-7B-Instruct | 79.2 | 49.5 | 74.8 | 41.1 | 37.2 | 9.8 | 34.5 | 8.5 | 76.5 | 72.5 | 84.0 | 81.2 | 46.5 | 19.5 | 52.8 | 24.2 |
| | Gemma3-12b-it | 81.5 | 57.8 | 73.5 | 30.9 | 44.5 | 9.4 | 33.8 | 8.2 | 80.2 | 75.2 | 83.1 | 79.8 | 45.2 | 18.8 | 51.5 | 23.1 |
| | InternVL3-14B | 85.4 | 63.6 | 78.1 | 36.2 | 47.7 | 10.5 | 35.2 | 9.1 | 83.2 | 81.2 | 85.4 | 83.5 | 48.5 | 21.2 | 54.8 | 26.5 |

**Text Generation.** Contextual captions were generated using *google/gemma-3-12b-it*. We used nucleus sampling with `temperature=0.7` and `top_p=0.9`. For all tasks, *max_new_tokens* was set to 150.

**Image Generation.** Images were generated using *stabilityai/stable-diffusion-3.5-large* at a 1024x1024 resolution. We used 28 inference steps with a guidance scale (CFG) of 4.5.

**Object Detection and Background Check.** For the object presence and count check, we used `IDEA-Research/grounding-dino-tiny` as the open-vocabulary detector. We set the `box_threshold=0.4` and `text_threshold=0.3`. The resulting bounding boxes were then used to prompt SAM2 (`sam2.1_hiera_large.pt`) for segmentation masks. For the subsequent background whiteness check on Minimal images, we used a grayscale brightness threshold of 190 and required at least 70% of non-object pixels to meet this threshold, a pragmatic choice to accommodate for minor shadows and lighting variations inherent to the generative process.

## A.6 PIPELINE GENERALIZABILITY AND MODEL SELECTION

The Auto-Comp pipeline is designed to be fully modular and model-agnostic. While the experiments in the main text utilize Gemma-3 and Stable Diffusion 3.5, these choices were not arbitrary. They were established through preliminary ablation studies aimed at maximizing benchmark quality (specifically caption diversity) and generation efficiency (sample survival rate).

### A.6.1 LLM SELECTION: OPTIMIZING FOR CAPTION DIVERSITY

We evaluated three state-of-the-art instruction-tuned models: Qwen2.5-VL-7B Bai et al. (2025), Llama-3.2-11B-Vision Grattafiori et al. (2024), and Gemma-3-12B-it Team et al. (2025). Since all models successfully adhered to our regex-based structural constraints, we selected the model based on Lexical Diversity (Distinct-N metrics) Li et al. (2016) and Semantic Diversity. The latter is calculated as $1 -$ Avg CLIP Score between generated captions, where higher values indicate greater semantic variance.

As shown in Table 15, Gemma-3 consistently produced the most diverse captions, achieving the highest Distinct-4 scores. While choosing a less diverse model would result in a more repetitive benchmark, the pipeline mechanics would remain functional.

Table 15: LLM Diversity Evaluation on Color and Position Benchmarks. Distinct-N measures n-gram diversity. Semantic Similarity represents semantic variance $(1 - \text{Avg CLIP Score})$.

| Model | Task | Complexity | Distinct-2 | Distinct-3 | Distinct-4 | Semantic Sim. (1-CLIP Score) |
|---|---|---|---|---|---|---|
| Qwen2.5-VL-7B-Instruct | Color | $N = 2$ | 0.27 | 0.48 | 0.66 | 0.61 |
| | | $N = 3$ | 0.29 | 0.52 | 0.71 | 0.60 |
| | Position | $N = 2$ | 0.24 | 0.44 | 0.62 | 0.57 |
| | | $N = 3$ | 0.27 | 0.49 | 0.65 | 0.54 |
| Llama-3.2-11B-Vision | Color | $N = 2$ | 0.30 | 0.54 | 0.73 | 0.66 |
| | | $N = 3$ | 0.33 | 0.58 | 0.78 | 0.64 |
| | Position | $N = 2$ | 0.27 | 0.49 | 0.68 | 0.61 |
| | | $N = 3$ | 0.30 | 0.54 | 0.70 | 0.58 |
| Gemma-3-12B-it | Color | $N = 2$ | 0.32 | 0.56 | 0.75 | 0.68 |
| | | $N = 3$ | 0.34 | 0.59 | 0.80 | 0.66 |
| | Position | $N = 2$ | 0.28 | 0.51 | 0.70 | 0.63 |
| | | $N = 3$ | 0.31 | 0.55 | 0.72 | 0.60 |

Table 16: T2I Model Faithfulness Evaluation. Higher CLIP and TIFA scores indicate better adherence to the prompt's constraints.

| Model | CLIPScore | TIFA |
|---|---|---|
| SD 2 Rombach et al. (2022) | 27.31 | 0.690 |
| SDXL Podell et al. (2023) Turbo | 30.75 | 0.879 |
| SD 3.5 Turbo Esser et al. (2024a) | 30.59 | 0.881 |
| SD 3.5 Large Esser et al. (2024a) | 32.79 | 0.914 |
| FLUX-schnell Labs et al. (2025) | 31.63 | 0.896 |
| FLUX-dev Labs et al. (2025) | 31.49 | 0.901 |

### A.6.2 T2I SELECTION: OPTIMIZING FOR FAITHFULNESS AND EFFICIENCY

For image generation, we evaluated models based on their ability to maintain coherence with the text prompt. This directly influences the "Sample Survival Rate" during the validation phase. We utilized TIFA (Text-to-Image Faithfulness Assessment), a VQA-based metric designed to measure fine-grained compliance, alongside standard CLIPScore.

As presented in Table 16, SD 3.5 Large achieved the highest TIFA score (0.914), outperforming SDXL, SD 2, and slightly exceeding FLUX-dev. High faithfulness is crucial for pipeline efficiency. Using a weaker model, such as SD 2, would not invalidate the pipeline due to our rigorous successive validation checks (GroundedSAM2 and VQA), which effectively filter out incorrect samples. However, a weaker generator significantly increases the computational costs required to obtain a benchmark of the same size due to higher rejection rates.

We also observe that distilled or "turbo" models offer an excellent trade-off: they maintain competitive text-image fidelity while drastically reducing inference steps, making them suitable for compute-constrained settings. Finally, utilizing larger and stronger future models would have a beneficial effect, further increasing survival rates and enhancing the cost-effectiveness of the pipeline.

### A.7 HUMAN VALIDATION DETAILS

To ensure the reliability of our generated benchmarks, we conducted a rigorous human performance study. We recruited four graduate-level evaluators who were blinded to the model generation process. Each evaluator assessed a balanced subset of 200 concepts (resulting in 400 images across Minimal and Contextual settings).

For every concept, evaluators performed two tasks:

1. **Swap Task:** Identifying the correct caption against a hard negative where attributes or objects were swapped.

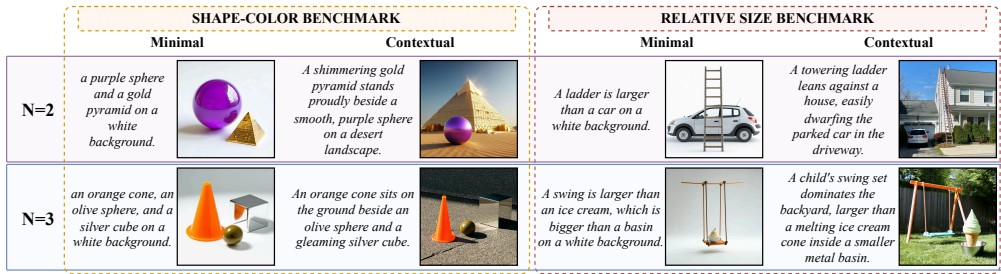

Figure 4: Qualitative examples from the newly generated *Shape-Color* and *Relative Size* benchmarks. The Auto-Comp pipeline generalizes to these new domains without architectural changes.

Table 17: **Per-Validator Human Performance.** Raw scores (out of 50) for each of the 4 independent human evaluators on the Swap and Confusion benchmarks. V1–V4 denote the four validators.

| Task | N | Condition | Swap Accuracy (/50) | | | | Confusion Accuracy (/50) | | | |
|------|---|-----------|-----|-----|-----|-----|-----|-----|-----|-----|
| | | | V1 | V2 | V3 | V4 | V1 | V2 | V3 | V4 |
| Color | $N = 2$ | Minimal | 50 | 49 | 48 | 48 | 49 | 49 | 49 | 48 |
| | | Contextual | 47 | 48 | 47 | 47 | 48 | 48 | 48 | 47 |
| | $N = 3$ | Minimal | 49 | 49 | 49 | 47 | 48 | 48 | 48 | 48 |
| | | Contextual | 48 | 47 | 47 | 47 | 47 | 47 | 47 | 47 |
| Position | $N = 2$ | Minimal | 49 | 49 | 49 | 48 | 48 | 49 | 48 | 48 |
| | | Contextual | 48 | 47 | 49 | 48 | 48 | 46 | 49 | 47 |
| | $N = 3$ | Minimal | 47 | 49 | 47 | 48 | 47 | 49 | 48 | 48 |
| | | Contextual | 46 | 47 | 47 | 47 | 48 | 47 | 46 | 47 |

2. **Confusion Task:** Identifying the correct caption against a set of low-entropy distractors (limited to 50 options for feasibility).

Table 17 reports the raw scores for each evaluator. The maximum possible score for each cell is 50. The results demonstrate high consistency across all four validators, with accuracies consistently exceeding 94% (47/50) for the vast majority of settings. Errors were qualitatively analyzed and found to be primarily driven by subtle color variations (e.g., distinguishing gold vs. yellow) or depth ambiguities in complex spatial scenes, rather than generation artifacts.

A.8    COMPUTATIONAL EFFICIENCY AND COST ANALYSIS

Auto-Comp is designed for high-throughput scalability. To quantify the computational resources required for benchmark generation, we conducted an efficiency analysis using a cluster of 16× NVIDIA A100 (80GB) GPUs. We leverage massive parallelism (batch size 16 per GPU) to maximize throughput during the generation phases.

Table 18 details the time required to process a batch of 100,000 initial concepts through the complete pipeline. It is important to note that the computational workload decreases significantly in the final stage. This is because the Object Detection step (Step 3) filters out approximately 53% of candidates (based on the average survival rates reported in Table 3 of the main text), drastically reducing the number of samples requiring VQA validation.

**Conclusion.**   Processing a batch of 100,000 concepts—which yields tens of thousands of high-quality validated samples depending on the specific task complexity—requires roughly **9.6 hours** on the cluster (approximately 154 GPU-hours total). This computational cost is orders of magnitude lower than the weeks of human labor required for manually curated benchmarks, demonstrating Auto-Comp's capability as a scalable data generation engine.

Table 18: **Pipeline Efficiency Analysis.** Estimated processing time for a batch of 100,000 initial concepts on a 16× A100 cluster. The VQA stage processes a reduced workload due to filtering in Step 3.

| Step | Task | Model Used | Workload | Est. Cluster Time (16× A100) |
|------|------|-----------|----------|------------------------------|
| 1 | Caption Gen | Gemma-3-12B-it | 100,000 prompts | ∼0.2 hours |
| 2 | Image Gen | SD 3.5 Large | 100,000 images | ∼8.5 hours |
| 3 | Obj/BG Check | Grounded-SAM-2 | 100,000 images | ∼0.8 hours |
| 4 | VQA Validation | Gemma-3-12B-it | ∼47,000 images[*] | ∼0.1 hours |
| **Total** | | | | **∼9.6 hours** |

[*]*Note: Only images passing object/background detection (Step 3) are forwarded to VQA validation.*

## A.9 LARGE LANGUAGE MODEL (LLM) USAGE

In adherence with conference policies, we report the use of a Large Language Model (LLM) in the preparation of this manuscript. Specifically, we utilized Google's Gemini 2.5 Pro as a collaborative partner to aid in polishing the writing. Its use was focused on improving the clarity, conciseness, and narrative flow of the text. All core scientific contributions, analyses, and results were conceived and generated by the human authors.

## A.10 CODE AND DATA AVAILABILITY

The source code for the Auto-Comp pipeline is provided in the supplementary material as a .zip file. The Auto-Comp-CP benchmark datasets, including all generated images and captions, are available for download at the following anonymous hf repo: `https://huggingface.co/AutoComp`. The Auto-Comp codebase, with generation, filtering and evaluation scripts is provided at the following anonymous repository: `https://github.com/iclrautocomp/AutoComp-Code`.

