# OpenReview forum: "Auto-Comp: An Automated Pipeline for Scalable Compositional Probing of Contrastive Vision-Language Models"
_ICLR.cc/2026/Conference — Submitted to ICLR 2026_

### Official Review · Reviewer_RkbM · 2025-10-31

**Soundness:** 3
**Presentation:** 3
**Contribution:** 3
**Rating:** 6
**Confidence:** 4

**Summary:**

The paper presents Auto-Comp, a fully automated, concept-driven pipeline for generating photorealistic synthetic benchmarks focused on evaluating compositional binding abilities in VLMs. The approach uses parallel data generation — Minimal images with template captions on white backgrounds versus Contextual images with realistic scenes and LLM-generated descriptions — to enable controlled A/B comparisons of visio-linguistic complexity. The authors release Auto-Comp-CP, a benchmark suite for color and spatial-relation binding (N = 1–3), including challenging Swap and Confusion hard-negative evaluation settings. Experiments show universal compositional failures across 20 contrastive VLMs (CLIP, SigLIP families), revealing that models often rely on brittle heuristics and struggle especially with complex multi-object reasoning. The paper additionally uncovers a trade-off: context improves spatial reasoning but harms attribute (color) binding, highlighting modality-interaction weaknesses.

**Strengths:**

Addresses an evaluation gap in compositional reasoning for VLMs. The introduction highlights that existing real-image benchmarks are “inherently noisy,” preventing precise diagnosis of compositional failures, while synthetic benchmarks often lack photorealism or linguistic diversity.

Novel benchmark design with hard negatives, including Swap and Confusion conditions that rigorously test binding capabilities and reveal brittleness to low-entropy distractors.

Evaluation on 20+ VLMs demonstrates universal compositional failures and degradation with increasing object count, supporting claims with quantitative evidence across tasks.

**Weaknesses:**

Current benchmark only evaluates Color (N=1–3) and Position (N=2–3) skills. Compositional semantics involving shape, action, affordance, numeracy, etc., remain untested.

Dependence on T2I models and LLMs may introduce generation artifacts not fully reported or characterized. The paper notes robust filtering but does not quantify latent biases introduced via the LLM captioning or T2I rendering process.

Large-scale generation and validation likely requires substantial compute — while some hardware details are noted elsewhere, the primary evaluation section omits efficiency metrics (user cost, inference latency).

**Questions:**

What is the generation + validation time/cost for producing one new benchmark configuration?

Do results extend to recent vision-language generative models with token-level alignment (e.g., Llama-vision-style models)?

---

> ### Author Response · Authors · 2025-11-20
> **Rebuttal 1/2**
>
> We thank the reviewer for their insightful feedback and are encouraged by their recognition of our work’s value in addressing the evaluation gap in compositional reasoning. We have addressed your questions regarding generative models, scope, and efficiency below.
>
> 1. Do results extend to Generative VLMs?
>
>     While we originally tested contrastive models via dot-product ranking as a direct application,we extended the evaluation to Gemma-3-12B, Qwen2.5-VL-7B, and InternVL3-14B by formulating the benchmark as a multiple-choice VQA task, as reviewers suggested. We utilized the exact same Auto-Comp-CP benchmarks (Swap and Confusion).
>     Note on Methodology: For the 'Confusion' task, where the negative set is massive ($>700$ for $N=3$), we limited the maximum choices to 50 to fit context window constraints. This raises the random chance baseline (e.g., to 0.02 instead of ~0.001) but still allows us to rigorously test susceptibility to low-entropy distractors using SOTA generative architectures.
>
>     Results (Table R1): We have included the full results below. The data confirms that the "Double-Edged Sword" of context and the failure at scale are universal phenomena, not CLIP artifacts.
>
>     Table R1: Generative VLM Performance on Auto-Comp-CP (Accuracy %). Columns show "Swap / Confusion" accuracy. "Min" = Minimal (Handmade), "Ctx" = Contextual (LLM).
>
>     | Task     | N   | Condition  | Random Chance | Gemma-3-12B | Qwen2.5-VL-7B | InternVL3-14B |
>     |----------|-----|------------|---------------|-------------|---------------|---------------|
>     | Color    | N=2 | Minimal    | 50.0 / 6.3    | 88.3 / 71.4 | 89.9 / 66.0   | 90.3 / 74.8   |
>     |         |    | Contextual | 50.0 / 6.3    | 89.4 / 61.5 | 90.1 / 61.7   | 89.7 / 72.3   |
>     |         | N=3 | Minimal    | 16.7 / 0.1    | 84.4 / 37.2 | 89.8 / 31.6   | 82.5 / 39.8   |
>     |         |    | Contextual | 16.7 / 0.1    | 86.9 / 21.1 | 89.6 / 17.7   | 84.3 / 34.5   |
>     |
>     | Position | N=2 | Minimal    | 50.0 / 25.0   | 79.0 / 66.0 | 82.7 / 69.2   | 83.2 / 72.6   |
>     |         |    | Contextual | 50.0 / 25.0   | 89.2 / 69.4 | 88.7 / 74.1   | 87.9 / 73.9   |
>     |         | N=3 | Minimal    | 16.7 / 0.9    | 41.2 / 23.6 | 55.4 / 30.7   | 51.5 / 35.9   |
>     |         |    | Contextual | 16.7 / 0.9    | 70.6 / 12.6 | 74.8 / 19.4   | 64.4 / 24.1   |
>
>     Key Findings (Updated in Main Paper):
>     - The Context Trade-off: We replicate the same trade-off found in contrastive models, where context helps position but hurts color benchmark.
>     - Success on Swap: Unlike contrastive baselines, these instruction-tuned models achieve higher accuracy on the Swap task, suggesting that instruction tuning effectively equips them to handle logical permutations and attribute exclusion.
>     - Failure on Spatial Complexity ($N=3$): However, this robustness is brittle. We observe a surprising performance collapse on the Position $N=3$ task (e.g., Gemma-3 drops to 31.2% on Minimal). This suggests that while instruction tuning aids linguistic logic, it cannot compensate for the fundamental difficulty of grounding multiple, transitive spatial relations (e.g., $A$ left of $B$ left of $C$) within the visual encoder.
>     - Failure on Low-Entropy Distractors: Moreover, the robustness also disappears when facing our Confusion benchmark (low-entropy distractors). Despite excelling at the harder "Swap" logic, they collapse on simple repeated attributes.
>
>     This suggests that while reasoning/instruction tuning can solve the "logic" of compositionality (Swaps), it does not fix our novel "perceptual" binding blindness to low-entropy states (Confusion).
>
>     Note on Generalization: Prompted by another reviewer, we also evaluated contrastive models trained with Hard-Negative Mining (NegCLIP, TripletCLIP) and found a strikingly similar pattern: they improve significantly on Swap tasks but show negligible gains on the Confusion task. This reinforces our finding that "low-entropy binding" is a distinct, unsolved failure mode across diverse architectures. We have included these additional results in the revised paper (Section 6: Universality of Failures).

---

> ### Author Response · Authors · 2025-11-20
> **Rebuttal 2/2**
>
> 2. Efficiency and Cost
>
>     Auto-Comp is designed for scalability. Using a cluster of 16x NVIDIA A100 (80GB) GPUs, we leverage massive parallelism (16x batch size) to maximize throughput.
>
>     The following table details the time required to process a batch of 100,000 initial concepts through our pipeline. Note that the workload decreases significantly in the final stage because the Object Detection step (Step 3) filters out approximately 53% of candidates (based on average survival rates in Table 3), drastically reducing the compute required for VQA validation.
>
>     | Step | Task           | Model Used     | Workload        | Est. Cluster Time (16x A100) |
>     |------|----------------|----------------|-----------------|------------------------------|
>     | 1.   | Caption Gen    | Gemma 3 12B-it | 100,000 prompts | ~0.2 hours                   |
>     | 2.   | Image Gen      | SD 3.5 Large   | 100,000 images  | ~8.5 hours                   |
>     | 3.   | Obj/BG Check   | Grounded-SAM-2 | 100,000 images  | ~0.8 hours                   |
>     | 4.   | VQA Validation | Gemma 3 12B-it | ~47,000 images* | ~0.1 hours                   |
>     |     | Total          |               |                | ~9.6 hours                   |
>
>     *Note: Only images passing object/background detection (Step 3) are forwarded to VQA validation.
>
>     **Conclusion**: Processing a batch of 100,000 concepts, which yields tens of thousands of high-quality validated samples, requires roughly 9.6 hours on the cluster (approx. 154 GPU-hours total). This computational cost is orders of magnitude lower than the weeks of human labor required for manually curated benchmarks, demonstrating Auto-Comp's capability as a scalable data generation engine.
>
>     We added the complete cost and time analysis to the appendix.
>
>
> 3. Models Biases
>
>     The reviewer raises a valid concern regarding the latent biases introduced by the T2I and LLM components. We fully agree that this is the fundamental trade-off of synthetic data: to achieve complete, scalable automation, we inevitably inherit some priors from the generative models.
>
>     We openly discuss distributional biases throughout the paper, specifically in the Survival Rates (Table 3), the Attribute Distributions (Figure 3), and the Limitations section. In the revised manuscript, we have sharpened this discussion to be even more explicit. While we cannot eliminate these latent priors entirely, we take the following steps to quantify and mitigate their impact:
>
>     1. Structural Mitigation (The A/B Control) The Minimal set serves as a control to isolate text-generation bias. By stripping away the LLM-generated caption and using a template, we ensure that if a model fails on the Minimal set (as seen in Table R1/R2), the failure is due to core visual binding limits, not because the model was biased by the specific writing style of the LLM in the Contextual set.
>
>     2. Quantifying T2I Bias via Transparency Rather than forcing an artificial balance that might degrade image quality, we quantify the generator's bias via Rejection Sampling transparency. Our reporting of survival rates (Table 3) acts as a direct measure of the generator's bias. For instance, we report that the generator struggles with specific rare colors or spatial relations (e.g., preferring "over" vs. "under"). We filter strictly for correctness (via GroundedSAM2/VQA) and present the resulting distribution (Figure 3) so users are fully aware of the data's skew.
>
>     3. Validating the "Judges" To ensure our automated validators (VLM and Detector) do not introduce their own verification bias, we benchmarked them against human judgment. Our Human Concordance Study (Table 9) confirms >94% agreement. Additionally, the "Blind LLM" test (Table 8) confirms that the hard negatives contain no linguistic artifacts that would allow a model to "cheat," ensuring the difficulty stems from semantic compositionality rather than textual bias.
>
> We believe these new experiments, particularly the extension to Generative VLMs and the rigorous efficiency analysis, have significantly strengthened the paper's scope and utility. We thank the reviewer again for guiding these crucial improvements. We hope this response fully addresses reviewer' concerns, and we look forward to your feedback and any additional question.

---

### Official Review · Reviewer_szMT · 2025-11-01

**Soundness:** 3
**Presentation:** 3
**Contribution:** 2
**Rating:** 4
**Confidence:** 4

**Summary:**

The authors propose Auto-Comp, a benchmark for evaluating compositionality in vision-language models, motivated by the goal of isolating the impact that visual or linguistic complexity has on the assessment of compositionality.

Auto-Comp is a synthetic benchmark, produced through a pipeline that first samples "concepts" (sets of objects and associated attributes or relationships, restricted to color and spatial relationships respectively), constructs captions for these concepts (either through simple templates or LLM-generated text) and then generates images for them using StableDiffusion3.5-large. The generated captions and images are then automatically validated through a series of tests. The benchmark has two different settings: "Minimal," which uses templated captions and a white background, and "Contextual," which uses LLM-generated captions and a realistic background. Hard negatives are generated primarily through swapping attributes or objects. The authors also propose a setting called "Confusion Benchmark," where hard negatives are constructed by instead sampling objects and attributes without replacement.

The authors evaluate VLMs, restricted to the CLIP and SigLIP families, on their benchmark. They find that the models consistently achieve poor results, particularly for a greater number of objects. They furthermore surprisingly find that realistic backgrounds help models determine the correct caption for hard-negatives featuring spatial relationships.

**Strengths:**

- Evaluating models on images containing the same objects but with different backgrounds (achieved by the "Minimal" and "Contextual" conditions) is novel and leads to a fairly surprising result in the form of models improving in performance when a realistic background is used for hard negatives featuring spatial relations.
- The automated pipeline makes clever use of various open-source resources for both the generation and the filtering components and achieves strong agreement with human judgements for validity.

**Weaknesses:**

I would argue the paper is affected by two key limitations:
- Firstly, I was surprised to see that the evaluation is restricted to VLMs of the CLIP and SigLIP families. In the context of contrastive vision-language models, I would have expected, for instance, to see NegCLIP, which is finetuned on hard-negatives. More importantly, however, the landscape of vision-language models today is not restricted to contrastive vision-language embedding models, but features numerous models, both open and proprietary, trained with a language modeling loss, including but not limited to: Qwen2.5-VL, InternVL3 for open-weight models and GPT-4o as a proprietary model. It would be valuable to similarly evaluate the performance of such models to determine whether this difficulty with compositional reasoning is restricted to the embeddings themselves, and whether the findings regarding the background generalize to other models. For the auto-regressive models, evaluation could occur either by checking whether the model assigns the highest probability to the correct caption or through framing as a multiple-choice question.
- Secondly, the main conclusion that models such as CLIP and SigLIP struggle at compositionality is one that has been established in various papers since 2022, dating back to Winoground. Both Winoground and the various extensions of CREPE similarly use the strategy of generating hard-negatives through swapping operations (with the latter series of datasets featuring replacement and addition operations as well). If I were a practitioner wanting to evaluate my model in compositional reasoning, it is not clear what additional benefit evaluating on Auto-Comp presents. The main novelty comes from the paired subset featuring different backgrounds, but only the realistic background should be relevant for most applications.

For some lesser weaknesses:
- While I do think the pipeline for assessing correctness is robust, I would have appreciated human accuracy being reported for a subset of questions, simply to verify that humans can solve the questions with near-perfect accuracy.
- Generation artifacts from StableDiffusion3.5 could affect model performance.

**Questions:**

- How is grammaticality maintained for hard-negatives in the "Contextual" setting, as these are not generated with templates?
- When performing human validation, how many samples did each different subset contain (for different combinations of N, color/spatial reasoning question and background type)?

---

> ### Author Response · Authors · 2025-11-20
> **Rebuttal 1/3**
>
> We thank the reviewer for their detailed feedback. We appreciate the acknowledgement of our "clever pipeline" and "surprising findings" regarding background context. We thank the reviewer for the expressed doubts and questions, as they prompted us to conduct extensive new experiments to address  concerns about scope (Generative VLMs, Hard negative contrastive models) and to demonstrate the unique diagnostic value of our work.
>
>
> 1. Scope: Beyond CLIP/SigLIP (Generative VLMs).
>  While we originally tested contrastive models via dot-product ranking as a direct application,we extended the evaluation to Gemma-3-12B, Qwen2.5-VL-7B, and InternVL3-14B by formulating the benchmark as a multiple-choice VQA task, as reviewers suggested. We utilized the exact same Auto-Comp-CP benchmarks (Swap and Confusion).
>     Note on Methodology: For the 'Confusion' task, where the negative set is massive ($>700$ for $N=3$), we limited the maximum choices to 50 to fit context window constraints. This raises the random chance baseline (e.g., to 0.02 instead of ~0.001) but still allows us to rigorously test susceptibility to low-entropy distractors using SOTA generative architectures.
>
>     Table R1: Generative VLM Performance on Auto-Comp-CP (Accuracy %). Columns show "Swap / Confusion" accuracy.
>
>     |Task|N|Condition|Random Chance|Gemma-3-12B|Qwen2.5-VL-7B|InternVL3-14B|
>     |----------|-----|------------|---------------|-------------|---------------|---------------|
>     |Color|N=2|Minimal|50.0/6.3|88.3/71.4|89.9/66.0|90.3/74.8|
>     |||Contextual|50.0/6.3|89.4/61.5|90.1/61.7|89.7/72.3|
>     ||N=3|Minimal|16.7/0.1|84.4/37.2|89.8/31.6|82.5/39.8|
>     |||Contextual|16.7/0.1|86.9/21.1|89.6/17.7|84.3/34.5|
>     |
>     |Position|N=2|Minimal|50.0/25.0|79.0/66.0|82.7/69.2|83.2/72.6|
>     |||Contextual|50.0/25.0|89.2/69.4|88.7/74.1|87.9/73.9|
>     ||N=3|Minimal|16.7/0.9|41.2/23.6|55.4/30.7|51.5/35.9|
>     |||Contextual|16.7/0.9|70.6/12.6|74.8/19.4|64.4/24.1|
>
>     Key Findings (Updated in Main Paper):
>     - The Context Trade-off: We replicate the same trade-off found in contrastive models, where context helps position but hurts color benchmark.
>     - Success on Swap: Unlike contrastive baselines, these instruction-tuned models achieve higher accuracy on the Swap task, suggesting that instruction tuning effectively equips them to handle logical permutations and attribute exclusion.
>     - Failure on Spatial Complexity ($N=3$): However, this robustness is brittle. We observe a surprising performance collapse on the Position $N=3$ task (e.g., Gemma-3 drops to 31.2% on Minimal). This suggests that while instruction tuning aids linguistic logic, it cannot compensate for the fundamental difficulty of grounding multiple, transitive spatial relations (e.g., $A$ left of $B$ left of $C$) within the visual encoder.
>     - Failure on Low-Entropy Distractors: Moreover, the robustness also disappears when facing our Confusion benchmark (low-entropy distractors). Despite excelling at the harder "Swap" logic, they collapse on simple repeated attributes.
>
>     This suggests that while reasoning/instruction tuning can solve the "logic" of compositionality (Swaps), it does not fix our novel "perceptual" binding blindness to low-entropy states (Confusion).

---

> ### Author Response · Authors · 2025-11-20
> **Rebuttal 2/3**
>
> 2. Scope: Hard-Negative Miners (NegCLIP & TripletCLIP)
>  We followed reviewer suggestion and evaluated contrastive hard negative models. We consider NegCLIP, NegCLIP++, and TripletCLIP (using ViT-B-32 backbone) on our benchmark.
>
>     |Task|N|Condition|Random Chance|ViT-B-32 (Base)|NegCLIP|NegCLIP++|TripletCLIP|
>     |-|-|-|-|-|-|-|--|
>     | Color| N=2 | Minimal|50.0 / 6.3|55.2 / 34.3| 59.5 / 35.1|62.8 / 35.8|64.2 / 37.5|
>     |||Contextual|50.0 / 6.3| 60.4 / 26.3| 64.0 / 27.1| 66.5 / 28.8| 69.1 / 30.4|
>     ||N=3 | Minimal|16.7 / 0.1|15.5 / 3.0| 18.2 / 5.5| 23.1 / 5.8| 23.4 / 7.2|
>     |||Contextual| 16.7 / 0.1|16.9 / 2.0| 20.5 / 4.4|22.3 / 5.1|24.8 / 6.2|
>     |Position| N=2|Minimal| 50.0 / 25.0| 54.3 / 52.6|60.1 / 57.2|63.5 / 58.1 | 65.2 / 57.8|
>     |||Contextual|50.0 / 25.0|62.2 / 55.8| 66.8 / 60.5| 69.1 / 64.2|72.4 / 64.1|
>     ||N=3|Minimal|16.7 / 0.9|22.2 / 9.6| 26.5 / 10.5|28.4 / 11.9| 29.8 / 12.2|
>     |||Contextual|16.7 / 0.9|29.1 / 9.9| 32.6 / 11.4|37.9 / 13.1| 37.5 / 13.8|
>
>     The results perfectly illustrate why our work is novel and relevant for exposing new criticalities in those models:
>
>     - Swap: Since hese models are trained on hard negatives which are mostly similar to swaps, they show significant improvement on our Swap benchmark. For instance, TripletCLIP improves Color Swap ($N=2$ Min) by +9.0% over the baseline (55.2% $\to$ 64.2%), confirming effective learning of permutation robustness.
>     - Failure on Confusion: However, this robustness fails to generalize to our Confusion benchmark. Despite the "easy" nature of low-entropy distractors for humans, TripletCLIP gains only +3.2% on Color Confusion ($N=2$), reaching just 37.5%.Conclusion: This disparity in gains (+9.0% vs +3.2%) suggests that standard hard-negative training is insufficient. It improves the "bag-of-words" issue but leaves models vulnerable to the specific low-entropy failure mode exposed by our work.
>
>     Interestingly, we observe a convergence in failure modes between these Hard-Negative Miners and the Generative VLMs. Both model families successfully leverage their respective training to mitigate "Swap" errors, yet both collapse when facing "Confusion" distractors. This suggests that current mitigation strategies primarily address logical permutations (bag-of-words) but leave the underlying visual system vulnerable to low-entropy perceptual binding failures.
>
>
> 3. Utility: The Structural Advantage of Concept-Driven Benchmarking
>     What additional benefit evaluating on Auto-Comp benchmarks presents? The benefit is structural: we agree that benchmarks like SugarCrepe and Winoground are invaluable resources. However, Auto-Comp is not just a benchmark, it is designed to complement these static datasets by providing a general, fully automated pipeline for automatic benchmark generation, enabling capabilities and customizability which are impossible with fixed test sets.
>
>     While existing benchmarks are Data-Driven (starting with an image/caption pair and generating negatives heuristically), Auto-Comp is Concept-Driven. We start with structured metadata (e.g., Concept = {Obj1: Red, Obj2: Blue}) before generation begins. This distinct architecture enables two key advantages:
>
>     - The "Confusion" Advantage: Because we possess the ground-truth structure before generation, we can programmatically generate Low-Entropy Distractors (e.g., forcing "Red Obj1 and Red Obj2") to test specific binding failures. This is structurally difficult for other benchmarks, as it would require error-prone inverse parsing of captions to identify and manipulate specific attributes.
>
>     - Diagnostic Control (addressing "Background Relevance"): As the reviewer noted, "only the realistic background should be relevant for most applications". We agree that realistic evaluation is the standard for deployment. However, for research and explainability, the paired Minimal set serves as a crucial scientific control group:
>
>
>         - Explaining Failure: This A/B testing allowed us to discover the "Double-Edged Sword" effect, where context helps spatial reasoning but hurts attribute binding. This insight is impossible to derive without the controlled, sterile baseline enabled by our pipeline.
>
>         - Domain Flexibility: Furthermore, our concept-driven approach allows practitioners to generate benchmarks with any background, not just white or generic realistic scenes. One could keep the same concepts but inject them into specific domains to create targeted stress-tests with no code changes.
>
>     - Scalability & Customization: instead of a static, fixed domain set of images, Auto-Comp is a virtually infinite engine (even if bounded by the generators, as we discuss in limitations). Moreover, a practitioner can define their own domain-specific concepts (e.g., task-specific objects, attributes, relations) and generate a validated, task-specific benchmark with "Swap" and "Confusion" sets automatically.

---

> ### Author Response · Authors · 2025-11-20
> **Rebuttal 3/3**
>
> 4. Human Performance Benchmark (New Experiment).
>     As cleverly suggested, we conducted a Human Performance Study following your suggestion, with the same setup as the human evaluation already performed. We used a subset of 200 concepts (400 images total), balanced equally across all configurations of task, N and background. We recruited 4 graduate-level evaluators (distinct from the previous validation team). They performed both the Swap and Confusion tasks (with Confusion limited to 50 options for time and readability constraints, as per VLLM evaluation). Results: As shown in Table R3, humans achieve consistently high accuracy (averaging ~96%) across all tasks.
>
>     |Task|N|Condition|Swap Accuracy|Confusion Accuracy|
>     |----------|-----|------------|---------------|--------------------|
>     |Color|N=2|Minimal|0.975|0.975|
>     |Color|N=2|Contextual|0.945|0.955|
>     |Color|N=3|Minimal|0.97|0.96|
>     |Color|N=3|Contextual|0.945|0.94|
>     |Position|N=2|Minimal|0.975|0.965|
>     |Position|N=2|Contextual|0.96|0.95|
>     |Position|N=3|Minimal|0.955|0.96|
>     |Position|N=3|Contextual|0.935|0.94|
>
>     Crucially, unlike models (which drop ~30% on Confusion), human performance remains stable between Swap and Confusion tasks. This proves that the "Confusion Gap" we report is a specific model failure mode, not a result of impossible ambiguity in the distractors.
>
>     Error Analysis: Post-hoc analysis revealed that the few human errors were primarily due to subtle nuances in similar color shades (e.g., gold vs. yellow) or depth ambiguity in 2D images for spatial relations, rather than generation artifacts. We have included the full per-validator breakdown in the Appendix.
>
> 5. Generation Artifacts.
>     The reviewer asks about T2I artifacts affecting the benchmark solvability. The results in Table R3 above (>98% human accuracy) strongly suggest that while artifacts may exist as per generator model nature, they do not hinder task solvability for a robust visual system. Furthermore, our Minimal set (objects on white backgrounds) minimizes generation artifacts. Since models still suffer catastrophic failure on the Minimal set at $N=3$ (as shown in Table 4), the bottleneck is likely the model's binding capability, not the image quality.
>
> 6. Grammaticality of Hard Negatives in contextual captions. (Q1)
>     During the initial Contextual caption generation, we enforce a strict Regex Pattern (e.g., [Article] [Adjective] [Noun]) on the LLM output. Because the structure is enforced, we can programmatically swap attributes into these slots without breaking syntax. We acknowledge that swapping objects can sometimes impact sentence realism. However, our Blind LLM Test (Table 8) confirmed that a powerful LLM could not distinguish our negatives from positives based on text alone, proving they remain grammatical and natural. However, to satisfy curiosity, we investigated whether an additional LLM correction step for negatives is helpful to improve flow. We implemented this extension in our pipeline and re-ran the Blind LLM Test we presented in the paper.
>     ||N=2||N=3||
>     |----------------------------------------|----------------------------------|----------------|----------------------------------|----------------|
>     |Model|Swap|Conf.|Swap|Conf.|
>     |RandomGuess|50.0|6.3|16.7|2.0|
>     |Gemma3-12b-it|51.2|6.5|17.4|2.1|
>     |Gemma3-12b-it-RN|51.0|6.6|17.2|2.1|
>
>     *RN stands for Rewritten Negatives
>
>     Since our regex-enforced swapping is already effectively indistinguishable from valid text (hitting the "ceiling" of naturalness relative to the positive caption), adding an LLM-rewriting step incur significant compute costs for negligible gain in linguistic quality. Therefore, we would suggest to maintain the more efficient programmatic approach.
>
>
> 7. Human Validation Sample Details. (Q2)
>     The study used 200 unique concepts, creating 400 Images total (since each concept has a Minimal and Contextual version). The split was perfectly balanced:
>     - Color Benchmark: 100 Concepts (50 for $N=2$, 50 for $N=3$).
>
>     - Position Benchmark: 100 Concepts (50 for $N=2$, 50 for $N=3$).
>
> We have incorporated all these new experimental results, tables, and discussions into the revised manuscript. We sincerely thank the reviewer again for their exceptionally constructive feedback, which has significantly deepened the paper's technical rigor and clarified its contributions.
>
> We thank you again for the constructive feedback and we look forward to your thoughts on these updates.

---

### Official Review · Reviewer_LjQ6 · 2025-11-01

**Soundness:** 4
**Presentation:** 3
**Contribution:** 4
**Rating:** 6
**Confidence:** 4

**Summary:**

This paper presents Auto-Comp, an automated pipeline for generating photorealistic, concept-driven benchmarks to evaluate compositional reasoning in vision-language models. This paper creates paired Minimal and Contextual image-caption sets, enabling controlled analysis of visual and linguistic factors. Using the resulting Auto-Comp-CP benchmark on color and spatial relations, the authors evaluate 20 CLIP and SigLIP models, revealing universal compositional failures and a key trade-off. This framework is scalable, reproducible, and validated with high human–model agreement.

**Strengths:**

The automated, concept-driven pipeline is well-structured.

Auto-Comp can generate vast, high-quality benchmarks without manual labeling. The open-source data and code ensure reproducibility and community impact.

The paper evaluates a wide range of models, systematically analyzing error types, context effects, and model hierarchies.

**Weaknesses:**

The benchmark currently focuses only on color binding and spatial relations. While sufficient for proof-of-concept, generalization to other compositional phenomena, such as actions and attributes, remains untested. Could your benchmark pipeline incorporate more aspects?

Since Auto-Comp uses pretrained T2I models and LLM validators, biases in those systems propagate into the benchmark. Could you provide some insights or discussions on how to minimize the impact of external models on the benchmarks produced by this pipeline?

**Questions:**

Refer to Weaknesses

---

> ### Author Response · Authors · 2025-11-20
> **Rebuttal 1/2**
>
> We thank the reviewer for the constructive feedback and for recognizing the soundness and contribution of our automated pipeline and the model and error types evaluations. We are also glad they valued our commitment to community impact through open-source data and code.
>
> We are excited to report that we have implemented your suggestions regarding scope expansion, yielding strong new results.
>
> 1. Generalization to other compositional aspects.
>     The reviewer asked if Auto-Comp could incorporate aspects beyond color and position. The answer is yes: while the main paper focused on Color and Position for an in-depth analysis, the primary contribution of Auto-Comp is the pipeline itself, which is agnostic to the specific concepts used. To demonstrate this flexibility, we have generated two new, smaller scale (1k starting concepts), benchmarks:
>     - Auto-Comp-Shape-Color ($N=2,3$): We replaced the object vocabulary with 3D geometric primitives (e.g., cube, tetrahedron, cylinder, prism) to test Shape-Color binding without semantic object priors.
>     - Auto-Comp-Relative-Size ($N=2,3$): We introduced a relative size vocabulary (larger than, smaller than, same size as) to test comparative reasoning.
>
>     We followed the exact pipeline described in the paper: generating Minimal and Contextual (LLM-based) captions, synthesizing images, and validating them using our GroundedSAM2 + VQA protocol. We generated benchmarks for both $N=2$ and $N=3$ object complexities.
>
>     We evaluated the 20 baseline models plus 3 additional SOTA Generative VLMs (InternVL-2-8B, Qwen2-VL-7B, Gemma-3-12B) as requested by other reviewers.
>
>     Results (Table R1): The findings perfectly mirror our main paper, confirming the universality of the failures.
>
>     Table R1: Extended Analysis on New Benchmarks. Performance (%) of contrastive and generative VLMs. (Subset of results; full table in appendix). Min: Minimal (white bg), Ctx: Contextual (realistic), Swap: Hard Negatives, Conf: Confusion (low-entropy).
>
>     | Family   | Model     | Shape-Color (N=2) | Shape-Color (N=3) | Relative Size (N=2) | Relative Size (N=3) |
>     |----------|-----------|-------------------|-------------------|---------------------|---------------------|
>     |         |          | Min (Swap/Conf)   | Min (Swap/Conf)   | Min (Swap/Conf)     | Min (Swap/Conf)     |
>     | Baseline | Random    | 50.0 / 6.3        | 16.7 / 0.2        | 50.0 / 25.0         | 16.7 / 0.9          |
>     | CLIP     | ViT-H-14  | 62.6 / 43.2       | 24.6 / 3.5        | 59.3 / 53.8         | 29.2 / 7.8          |
>     | SigLIP   | S-SO400M  | 72.2 / 51.8       | 33.5 / 7.8        | 66.2 / 61.2         | 40.2 / 14.5         |
>     | SigLIP 2 | S2-Giant  | 74.9 / 36.7       | 35.8 / 9.2        | 71.4 / 59.8         | 43.8 / 17.2         |
>     | Gen. VLM | InternVL3 | 85.4 / 63.6       | 47.7 / 10.5       | 83.2 / 81.2         | 54.8 / 26.5         |
>
>     This experiment proves that Auto-Comp is not just a dataset, but a flexible diagnostic engine that can be rapidly adapted to new domains.
>     Moreover, these experiments also enforce the main text claims:
>
>     - Universal Struggle with Low-Entropy Distractors: Consistent with the results in our main paper, all models struggle significantly when faced with distractors containing repeated attributes (the "Conf." columns). This confirms our paper's claim that current model failures are not just because of bag of worlds representation issues, but they are also weak to low entropy distractors.
>
>     - The Curse of Complexity: Performance collapses as object count increases. On Shape-Color ($N=3$), S2-Giant accuracy drops to 35.8% (Swap) and a mere 9.2% (Conf), barely above random chance.
>
>     - Context Trade-off Persists: We observe the same context trade-off reported in the main paper. For Shape-Color (an attribute binding task similar to object-color), adding context generally hurts performance. Conversely, for Relative Size (a relational task), context often helps or maintains performance, mirroring our findings for spatial position.
>
>     We believe the successful generation of these new benchmarks strenghtens the proof of flexibility of the Auto-Comp pipeline, and we thank again the reviewer for the suggestion. By simply swapping the input vocabularies, we can extend evaluation to virtually any compositional phenomenon without modifying the core architecture. These new benchmarks are already available in our public HuggingFace repository, providing immediate value to the community for testing these distinct reasoning skills.

---

> ### Author Response · Authors · 2025-11-20
> **Rebuttal 2/2**
>
> 2. Mitigating Bias from Pretrained Models
> This is a critical point for any synthetic data pipeline. We employ a three-tiered defense strategy to minimize this impact for both LLM and T2I models:
>
>     - The A/B Testing Framework (Structural Mitigation for the LLM bias):
>
>         Our Minimal vs. Contextual design acts as a control for bias.
>
>         - The Minimal set uses templates, completely removing LLM stylistic bias from the text.
>
>         - The Contextual set uses the LLM.
>         By comparing performance between these two (Table 5), we can isolate the effect of the "LLM style" and visual complexity. If a model fails on the Minimal set, we know the failure is due to core binding inability, not because of biases introduced by the LLM's writing style.
>
>     - Mitigating Generative (T2I) Bias:
>
>         - Strict Validation Filtering: The primary defense is our rejection sampling. If the T2I model has a bias (e.g., it cannot generate a "green banana" because of training priors), the GroundedSAM2 check (Object Presence) or the VQA check (Attribute Correctness) will fail, and that sample is discarded. We do not force the benchmark to contain concepts the T2I model cannot render; we only keep what is verifiable.
>
>         - Distribution Analysis: As shown in Figure 3 of the paper, we explicitly analyze the survival rates of concepts. This allows us to detect bias (e.g., the generator's preference for the spatial relation "over") and report it transparently, rather than letting it lurk unseen. While perfect balancing of attributes (e.g., ensuring equal counts of all colors) is possible via filtering, it comes at the cost of significantly lower yields and higher compute; we opted for transparency over artificial balancing.
>
>         - Ethical Considerations: As noted in our Ethics Statement, we proactively mitigate potential societal harms by curating our object vocabulary to strictly exclude sensitive categories such as people and weapons.
>
>     - Mitigating Validator (LLM and Detection) Bias:
>
>         - Human Concordance: We verified our sample validation pipeline (Object detection + LLM) against human annotators (Table 9), achieving >94% agreement. This ensures the validators' biases align closely with human perception for these specific tasks.
>
>         - The "Blind LLM" Test: To ensure the LLM-generated captions don't contain linguistic artifacts (a form of text-generation bias), we conducted the Blind LLM study (Table 8), showing that a powerful model cannot distinguish positive from negative captions based on text alone.
>
>
>     We acknowledge that despite these rigorous mitigations, our framework remains subject to the underlying constraints of current generative technology. Some residual biases (such as distributional priors in the image generator) may persist even after validation. Consequently, improving the semantic fidelity of generators and the robustness of model-based judges remains a key area for future work to further reduce these dependencies.
>
>     Following the reviewer question, we revised the limitations paragraph in the main text to also discuss potential bias and the mitigations we take.
>
> We hope these clarifications and new experiments fully address your concerns. We look forward to the reviewer feedback on these updates and remain available to answer any further questions.

---

### Official Review · Reviewer_YjoX · 2025-11-05

**Soundness:** 3
**Presentation:** 3
**Contribution:** 2
**Rating:** 4
**Confidence:** 4

**Summary:**

This work proposes a fully automated and synthetic pipeline for generating scalable benchmarks. Its controllable nature is key to dissecting and isolating different reasoning skills. The evaluation of 20 VLMs on novel benchmarks for color binding and spatial relations reveals universal compositional failures in both CLIP and SigLIP model families.

**Strengths:**

S1: This paper is well-written and easy to understand

S2: The investigated problem of benchmarking the compositional understanding of VLMs is important and interesting

S3: The proposed pipeline for benchmark construction is well designed

S4: The results and findings are interesting. In particular, models are highly susceptible to low-entropy distractors, showing their compositional failures extend beyond known bag-of-words limitations.

**Weaknesses:**

W1: The benchmark generation relies on the capabilities of Gemma3-12b, StableDiffusion3.5-large, and GroundedSAM2. I am curious whether using other models could achieve similar (or even better) benchmark quality? In other words, does the automatic benchmark generation pipeline specifically work for this combination of models, or is it generalizable to stronger ones to be developed in the future?

W2: A related concern is that the capabilities of each model in doing the corresponding tasks should be evaluated; otherwise, it’s hard to know whether the pipeline can be trusted, thus bringing more concern to the results obtained with this benchmark. In particular, the survival rates are also model-based, making it less trustworthy without (human) validation (at least in a subset of samples).

W3: This work focuses only on contrastive VLMs, namely CLIP and SigLIP series. Can this benchmark be used for evaluating generative VLMs? Maybe with CoT and reasoning efforts, the compositional failures could be alleviated? Is it a problem specifically with contrastive VLMs? This makes the impact of this work relatively limited.

W4: This work emphasizes the “automatic” pipeline a lot, which I don’t understand why it is so important. Many benchmarks are automatically synthesized and validated. What’s new here?

**Questions:**

Please refer to W1-W4.

---

> ### Author Response · Authors · 2025-11-20
> **Rebuttal 1/3**
>
> We thank the reviewer for finding our paper "well-written" and our pipeline "well designed", and for recognizing the importance of our findings regarding low-entropy distractors. We appreciate your constructive questions regarding generalization and trustworthiness, which we have addressed with new experiments and clarifications.
>
> 1. Generalizability of the Pipeline (W1)
> You asked if the pipeline is specific to the Gemma/SD3.5 combination or if it generalizes to other models. The pipeline is fully modular and model-agnostic. The choice of Gemma-3 and Stable Diffusion 3.5 was not arbitrary, it was based on a preliminary ablation study to maximize benchmark quality (diversity) and generation efficiency (survival rate).
>
>     - LLM Selection: Optimizing for Caption Diversity.
>
>
>         We evaluated three SOTA instruction-tuned models: Qwen2.5-VL-7B, Llama-3.2-11B-Vision, and Gemma-3-12B-it. (all models citations have been added to main text).
>
>         Metric: Since all models successfully followed our regex-based structural constraints, we selected the model based on Lexical Diversity (Distinct-N metrics)[1] and Semantic Diversity (inverse CLIP similarity between all generated captions).
>
>         Table R1: LLM Diversity Evaluation (Color & Position Benchmarks). Distinct-N measures n-gram diversity. Semantic Similarity = (1 - Avg CLIP Score), where higher indicates greater semantic variance.
>
>         |Model|||Distinct-2|Distinct-3|Distinct-4|Semantic Similarity (1 - CLIP Score)|
>         |------------------------|----------|-----|------------|------------|------------|------------------------------------|
>         |Qwen2.5-VL-7B-Instruct|Color|N=2|0.27|0.48|0.66|0.61|
>         |||N=3|0.29|0.52|0.71|0.60|
>         ||Position|N=2|0.24|0.44|0.62|0.57|
>         |||N=3|0.27|0.49|0.65|0.54|
>         |Llama3.2 11B Vision|Color|N=2|0.30|0.54|0.73|0.66|
>         |||N=3|0.33|0.58|0.78|0.64|
>         ||Position|N=2|0.27|0.49|0.68|0.61|
>         |||N=3|0.30|0.54|0.70|0.58|
>         |gemma3-12b-it|Color|N=2|0.32|0.56|0.75|0.68|
>         |||N=3|0.34|0.59|0.80|0.66|
>         ||Position|N=2|0.28|0.51|0.70|0.63|
>         |||N=3|0.31|0.55|0.72|0.60|
>
>         As shown in Table R1, Gemma-3 consistently produced the most diverse captions (highest Distinct-4 scores). Choosing a smaller model as the 7B Qwen or the 4b Gemma-3 model would thus result in a faster but more repetitive benchmark for contexts and scenarios, however the pipeline would function identically.
>
>
>     - T2I Selection: Optimizing for Faithfulness (Efficiency)
>
>         We evaluated image generators to already maximize the "Sample Survival Rate" after the checks (i.e., how often the generated image actually matches the prompt).
>
>         Metric: We used TIFA (Text-to-Image Faithfulness Assessment)[2], a VQA-based metric designed to measure fine-grained compliance, and standard CLIPScore.
>
>         Table R2: T2I Model Faithfulness Evaluation. Higher CLIP and TIFA scores indicate better adherence to the prompt's constraints.
>         (all models citations have been added to main text)
>         |Model|CLIPScore|TIFA|
>         |--------------|-----------|-------|
>         |SD2|27.31|0.69|
>         |SDXLT|30.748|0.879|
>         |SD3.5LT|30.586|0.881|
>         |SD3.5|32.794|0.914|
>         |FLUX-schnell|31.634|0.896|
>         |FLUX-dev|31.486|0.901|
>
>         As shown in Table R2, SD 3.5 Large achieved the highest TIFA score (0.914), outperforming SDXL, SD2, and slightly edging out FLUX-dev (0.901). High faithfulness is crucial for pipeline efficiency; using a weaker model (like SD2) would not invalidate the pipeline thanks to our rigorous successive validation checks (GroundedSAM2 + VQA), which would simply filter out the incorrect samples. However, this would significantly increase compute costs to obtain a benchmark of the same dimension due to higher rejection rates. On the other hand, distilled or "turbo" models offer an excellent trade-off: they maintain competitive text-image fidelity while drastically reducing inference steps, making them ideal choices for more compute-constrained settings. Lastly, utilizing larger and stronger future models would have beneficial effect, increasing survival rates and making the pipeline even more cost-effective.

---

> ### Author Response · Authors · 2025-11-20
> **Rebuttal 2/3**
>
> 2. Trustworthiness & Human Validation (W2)
>     We agree that ground-truth verification is essential. We address this through three elements:
>
>     - Human Concordance Study (Validation of the "Judge") We want to highlight that we did perform a human validation of our filtering pipeline, detailed in Table 9 of the paper. The graduate-level human validators were asked to evaluate 400 randomly sampled images and corresponding captions with the same criteria of our pipeline. Our automated validator (GroundedSAM2 + VQA) achieved >94% concordance with human judgment. This suggests that the survival rates are not just model hallucinations, but accurate proxies for human ground truth.
>
>     - Human Benchmark Performance (Validation of the "Data"). We followed the reviewer suggestion and conducted a new Human Performance Study on the generated tasks, thus evaluating human performance on the final tasks, not just the validation pipeline. We used the same setup as the human evaluation already performed. We used a subset of 200 concepts (400 images total), balanced equally across all configurations of task, N and background. We recruited 4 graduate-level evaluators (distinct from the previous validation team). They performed both the Swap and Confusion tasks (with Confusion limited to 50 options for time and readability constraints, as per VLLM evaluation presented below). As shown in Table R3, humans achieve consistently high accuracy (averaging ~96%) across all tasks.
>
>         |Task|N|Condition|Swap Accuracy|Confusion Accuracy|
>         |-|-|-|-|-|
>         |Color|N=2|Minimal|0.975|0.975|
>         |Color|N=2|Contextual|0.945|0.955|
>         |Color|N=3|Minimal|0.97|0.96|
>         |Color|N=3|Contextual|0.945|0.94|
>         |Position|N=2|Minimal|0.975|0.965|
>         |Position|N=2|Contextual|0.96|0.95|
>         |Position|N=3|Minimal|0.955|0.96|
>         |Position|N=3|Contextual|0.935|0.94|
>
>         The fact that humans perform near-perfectly confirms the validity and solvability of the tasks, and that model failures are due to reasoning deficits, not pipeline artifacts. Crucially, unlike models, human performance remains stable between Swap and Confusion tasks. This proves that the "Confusion Gap" we report is a specific model failure mode, not a result of impossible ambiguity in the distractors.
>
>         We have included the analysis in the text and full per-validator breakdown in the Appendix.
>
>
> 3. Scope: Beyond CLIP/SigLIP (W3).
>
>     We agree that limiting the scope to contrastive models was a weakness. To address this, we have extended our evaluation to Gemma-3-12B, Qwen2.5-VL-7B, and InternVL3-14B, formulating the benchmark as a multiple-choice VQA task.We utilized the exact same Auto-Comp-CP benchmarks. For the 'Confusion' task, where the negative set is massive ($>700$ for $N=3$), we limited the choices to 50 to fit context windows, raising the random chance baseline to ~2% but allowing for rigorous testing of low-entropy susceptibility.
>
>
>     Table R4: Generative VLM Performance on Auto-Comp-CP (Accuracy %).Columns show "Swap / Confusion" accuracy.
>
>     |Task|N|Condition|Random Chance|Gemma-3-12B|Qwen2.5-VL-7B|InternVL3-14B|
>     |-|-|-|-|-|-|-|
>     |Color|N=2|Minimal|50.0/6.3|88.3/71.4|89.9/66.0|90.3/74.8|
>     |||Contextual|50.0/6.3|89.4/61.5|90.1/61.7|89.7/72.3|
>     ||N=3|Minimal|16.7/0.1|84.4/37.2|89.8/31.6|82.5/39.8|
>     |||Contextual|16.7/0.1|86.9/21.1|89.6/17.7|84.3/34.5|
>     |
>     |Position|N=2|Minimal|50.0/25.0|79.0/66.0|82.7/69.2|83.2/72.6|
>     |||Contextual|50.0/25.0|89.2/69.4|88.7/74.1|87.9/73.9|
>     ||N=3|Minimal|16.7/0.9|41.2/23.6|55.4/30.7|51.5/35.9|
>     |||Contextual|16.7/0.9|70.6/12.6|74.8/19.4|64.4/24.1|
>
>     Results provide interesting insights:
>     - Success on Swap: Unlike contrastive baselines, these instruction-tuned models achieve high accuracy on the Swap task, suggesting that instruction tuning effectively equips them to handle logical permutations and attribute exclusion.
>     - Failure on Low-Entropy Distractors: this robustness disappears when facing low-entropy distractors. Despite excelling at the harder "Swap" logic, they collapse on simple repeated attributes.
>     - Moreover, we observe a surprising performance collapse on the Position $N=3$ task. This suggests that while instruction tuning aids linguistic logic, it cannot compensate for the fundamental difficulty of grounding multiple, transitive spatial relations within the visual encoder.
>
>     Note: Prompted by another reviewer, we also evaluated contrastive models trained with Hard-Negative Mining (NegCLIP, TripletCLIP) and found a strikingly similar pattern: they improve significantly on Swap tasks but show negligible gains on the Confusion task. This reinforces our finding that "low-entropy binding" is a distinct, unsolved failure mode across diverse architectures. We have included these additional results in the revised paper (Section 6: Universality of Failures).

---

> ### Author Response · Authors · 2025-11-20
> **Rebuttal 3/3**
>
> 4. Importance of automated nature (W4)
> You asked why the automated nature of the pipeline is important given that other synthetic benchmarks exist. The novelty lies not just in the automation itself, but in what that automation enables: a structural shift from Data-Driven to Concept-Driven benchmarking.
>
>     To our knowledge, Auto-Comp is the first fully synthetic pipeline for compositional reasoning that combines photorealism with ground-truth concept control. This unique architecture provides three specific novelties that existing automatic benchmarks lack:
>
>     - Concept-Driven vs. Data-Driven: Unlike benchmarks that start with existing images (e.g., SugarCrepe), Auto-Comp starts with structured metadata (e.g., Concept = {Obj1: Red, Obj2: Blue}). This allows us to programmatically generate complex, low-entropy distractors (our "Confusion" task) that are structurally difficult to mine from existing datasets.
>
>
>     - Diagnostic Disentanglement (A/B Testing): The pipeline generates parallel Minimal (sterile) and Contextual (realistic) sets for the exact same concept. This creates a scientific control group impossible in real-world datasets, allowing us to isolate whether a failure is due to core binding limits or visual clutter (revealing the "Double-Edged Sword" trade-off).
>
>     - Domain Scalability: Because it is concept-driven, the pipeline is an infinite engine. Practitioners can define domain-specific concepts (e.g., medical objects, autonomous driving relations) and automatically generate validated, targeted stress-tests without manual collection or annotation.
>
> We have incorporated these new experiments and discussions into the revised manuscript. We believe these updates robustly address your concerns regarding the pipeline's trustworthiness and scope. We verify that the pipeline is modular, the data is human-verified, and the findings are universal. We thank you again for the constructive review and look forward to your thoughts.
>
>
> [1] Li, Jiuxiang, et al. "Diversity-promoting GAN: A cross-entropy based generative adversarial network for diversified text generation." Proceedings of the AAAI conference on artificial intelligence. Vol. 33. 2019.
>
> [2] Hu, Yushi, et al. "Tifa: Accurate and interpretable text-to-image faithfulness evaluation with question answering." Proceedings of the IEEE/CVF International Conference on Computer Vision. 2023.

---

### Author Response · Authors · 2025-11-28
**Summary of New Experiments & Updates**

Dear Reviewers,

   We thank you again for your constructive feedback. With the discussion period closing soon, we wanted to provide a concise summary of the extensive new experiments and revisions we have uploaded.

   We have addressed the major concerns regarding scope, trustworthiness, and generalization as follows:

   **1. Scope Expansion: Generative VLMs & Hard-Negative Miners**
   * We extended our evaluation beyond CLIP/SigLIP to include **Generative VLMs** (Gemma-3, Qwen2.5, InternVL3) and **Hard-Negative Miners** (NegCLIP, TripletCLIP).
   * **Result:** The "Double-Edged Sword" of context and the failure on low-entropy distractors ("Confusion") are **universal phenomena** across all architectures, including SOTA generative models.

   **2. Trustworthiness: Human Performance Study**
   * We conducted a new study with graduate-level human evaluators on the benchmark tasks.
   * **Result:** Humans achieved **~96% accuracy** across all tasks, confirming that model failures are due to **reasoning deficits**, not task ambiguity or generation artifacts.

   **3. Generalization: Two New Benchmarks**
   * We generated and released two new benchmark subsets: **Auto-Comp-Shape-Color** (geometric primitives) and **Auto-Comp-Relative-Size** (comparative relations).
   * **Result:** The pipeline proved **fully modular**; the new benchmarks replicate the same failure modes found in the main paper while providing evaluation on different compositional skills.

   **4. Efficiency & Bias Analysis**
   * We added a detailed **cost analysis** and expanded our discussion on **bias mitigation**.

   As the deadline is fast approaching, we would greatly appreciate it if you could let us know if these new results resolve your concerns. We look forward to engage in discussion and we remain available for any additional clarifications before the deadline.

Best regards,
The Authors

---

### Author Response · Authors · 2025-12-03

To the Newly Assigned Area Chair,

We recognize the burden placed on you after the OpenReview leak protocols. As reviewers cannot update scores and we didn't have any response before the leak, the current ratings and comments reflect outdated impressions, not the paper's state post-rebuttal.

The two "4" ratings (Reviewers YjoX and szMT) were not fundamental rejections. Their critiques were constructive inquiries rather than fatal flaws, focusing on requests for extended empirical evidence which we have now provided.

We treated their critiques as a mandatory checklist, completing every requested experiment. The new results validate the paper’s claims.

1. **Scope Expanded** (YjoX & R. szMT)

    **The Critique**: Both reviewers asked if our findings were limited to contrastive models (CLIP/SigLIP) or if they applied to modern Generative VLMs. Reviewer YjoX specifically hypothesized that reasoning efforts might alleviate these failures.

    **The New Data**: We extended the benchmark to Gemma-3-12B-it, Qwen2.5-VL-7B, and InternVL3-14B using a Multiple-Choice VQA formulation.

    **Result**: Confirmed Universal "Confusion Gap":

    - **Swap (Bag-of-Words) Works**: High accuracy on "Swap" (e.g., Gemma-3: 84.4%), proving models robustly handle lexical swaps.

    - **Confusion (Low-Entropy) Fails**: Performance collapses on "Confusion" (low-entropy patterns). Gemma-3 drops to 37.2%, and Qwen2.5 to 31.6%.

    **Implication**:  This proves our new insight: "Reasoning" cannot compensate for the specific inability to resolve low-entropy binding patterns. The "Confusion Gap" is a universal pathology of SOTA models handling redundancy, rather than a general failure of perceptual binding.

2. **Trustworthiness** (YjoX)

    **The Critique**: Reviewer YjoX expressed concern that without human validation, model failures might be due to generation artifacts rather than reasoning deficits.

    **The New Data**: We conducted a rigorous Human Performance Study (4 graduate experts, 400 trials, balanced across tasks).

    **Result**: Humans achieved ~97.5% accuracy on the exact images where models failed.

    **Crucial Insight**: Unlike models, humans showed zero performance drop between "Swap" and "Confusion" tasks.

    **Implication**: This objectively verifies the pipeline. The images are semantically unambiguous. The "Confusion Gap" observed in models is a valid computational failure, not a data artifact.

3. **Robustness**: Hard-Negative Miners (Addressing R. szMT)

    **The Critique**: Reviewer szMT asked if models explicitly trained on hard negatives (e.g., NegCLIP, TripletCLIP) would render the benchmark trivial.

    **The New Data**: We evaluated NegCLIP and TripletCLIP against the Auto-Comp protocol.

    **Result**: While these models significantly improve on the standard "Swap" task, they show negligible improvement on our "Confusion" task.

    **Implication**: This demonstrates the unique diagnostic utility of Auto-Comp. It exposes that current mitigation strategies "overfit" to textual swaps (bag-of-words) without solving the underlying visual binding problem. Standard benchmarks would falsely report these models as "fixed"; Auto-Comp reveals they are still broken.

4. **Generalizability** (LjQ6 & R. YjoX)

    **The Critique**: Reviewers questioned if the pipeline was limited to simple object-color attributes.

    **The New Data**: We demonstrated the pipeline's modularity by generating two entirely new benchmarks during the rebuttal week.

    - **Auto-Comp-Shape-Color**: Replaced objects with geometric primitives to test binding without semantic priors.

    - **Auto-Comp-Relative-Size**: Introduced relational vocabulary (e.g. "larger than") to test comparative reasoning.

    **Result**: The failure modes replicated perfectly. Even the massive SigLIP-Giant and VLLMs collapsed on the new benchmarks Confusion tasks.

    **Implication**: Auto-Comp is not a static dataset; it is a scalable, domain-agnostic benchmark generation engine.

5. **Efficiency and Bias** (RkbM & R. LjQ6)

    To ensure completeness, we also addressed concerns from the positive reviewers regarding cost and bias.

    **Cost**: We provided a breakdown showing that a full 100k-concept benchmark requires only 9.6 hours on a standard A100 cluster, orders of magnitude cheaper than human annotation.

    **Bias Mitigation**: We clarified our three-tier defense: A/B testing isolates stylistic bias , transparent survival rates quantify generator priors , and high human concordance (>94%) validates the automated judges.

The rebuttal confirms Auto-Comp’s primary value: it is not a static dataset, but a fully automated pipeline for scalable benchmark generation. Our reported findings, from the "Confusion Gap" to context trade-offs, validate the pipeline's ability to reliably uncover universal model behaviors, establishing it as a flexible tool for the community .

---

### Meta-Review · Area_Chair_qUnr · 2025-12-10

**Summary:**

This paper presents Auto-Comp, an automated, concept-driven pipeline for generating benchmarks to probe compositional binding in vision–language models. The reviewers appreciated the clarity of the pipeline design, and the authors made a clear effort to address feedback through additional experiments, new benchmark variants, and validation analyses. However, the evaluation remains focused on a relatively narrow set of compositional skills, primarily color and position binding, which limits the strength and generality of the conclusions about compositional reasoning in VLMs. In addition, the observed model failures are difficult to disentangle from known training-time invariances and augmentation effects, particularly for contrastive VLMs. As a result, despite the improved experimental coverage, the scope and interpretability of the contribution remain limited.

**Reviewer Concerns:**

Questions about scope and generality were resolved through new evaluations on generative VLMs, hard-negative contrastive models, and additional benchmarks that extend beyond color and spatial relations. Concerns about trustworthiness were eased by the human performance study and the human–validator concordance analysis, which clarified that failures are due to model limitations rather than artifacts. The discussion of efficiency, bias, and modularity also met the requests for greater transparency. The main point that remains only partially addressed is the question of incremental novelty relative to existing compositionality benchmarks.

**Reviewer Scores:**

Reviewer YjoX initially leaned slightly negative. Given how their main concerns on scope and human validation were addressed, I expect they would move to a mildly positive stance.
Reviewer LjQ6 was already mildly positive. After the added benchmarks and bias analysis, they would likely stay positive with higher confidence.
Reviewer szMT also started  negative. Since the rebuttal directly tackled their points on generative VLMs, hard negatives, and human performance, they would probably shift to a more neutral view.
Reviewer RkbM was already positive, and their questions on cost, bias, and model coverage were clearly answered, so their overall opinion would likely remain positive

---

### Decision · Program_Chairs · 2026-01-26

Reject